



# Model development for simulating mudslide and the case study of the failure of the gypsum tailings dam in East Texas in 1966

Tso-Ren Wu [1*], Thi-Hong-Nhi Vuong [1], Chun-Yu Wang [2], Chia-Ren Chu [2], Chun-Wei Lin[1]

[1] Graduate Institute of Hydrological and Oceanic Sciences, National Central University, Taiwan
5 [2] Department of Civil Engineering, National Central University, Taiwan

*Correspondence to*: Tso-Ren Wu (tsoren@ncu.edu.tw)

**Abstract.** Mudslides, avalanches, and mine dam-breaks can be serious disasters and cause severe damages but the detailed flow field description has not been completed yet. This paper developed a modified Bi-viscosity model (MBM) to solve the mudslide flow by adopting Bingham model (BM) and the conventional Bi-viscosity model (CBM). In both CBM and MBM, 10 a yield strain rate is used to identify the plug and liquefied rheological prosperities. In the MBM, an extremely high plug viscosity adopted to represent the stratification effect. BM, CBM, and MBM are integrated into the Splash3D model, which solves Navier-Stokes equations with PLIC-VOF surface-tracking algorithm. The viscosity term is solved by implicit iteration. The model is carefully validated with theoretical results and laboratory data with good agreements. The Splash3D model is then used to study the failure of the gypsum tailings dam in East Texas in 1966, briefed as 'FGT66'. A series of sensitivity 15 analyses on the yield strain rate and grid resolution is performed. The results show that the predicted flood distance and flood speed by MBM is very close to the field data. The MBM results illustrate the process that the plug zone and liquefied zone is developed. The simulations show the initiation of the mudslide flow, the development of the slip surface, the flooding process, and the velocity ceasing process. The slip surface is developed automatically without empirical equations. By comparing the results of BM, CBM, and MBM to the field data, we conclude that the liquefied tailings are under the effect of stratification, 20 and the stratification effect is presented in the extremely high plug viscosity in the Splash3D model.

## 1 Introduction

According to the statistics of the World Information Service on Energy (WISE), more than 128 tailings dam failures occurred from 1961 to 2019 in the world. Failure of tailings dam causes loss of lives, irreversible damages to ecosystems and large economic damages. It, therefore, is necessary to study tailings dam risk seriously, especially in the numerical study. A model 25 can propose some scenarios helping scientists make contingency plans before the worst scenarios happen.

Many tailings dam events have been reported in the literature, but unfortunately, the information is not complete. It is difficult to obtain topography data, and on many occasions only approximate one-dimensional profiles are available. Properties of the flowing material are not only scarce but also contradictory (Pastor et al., 2002).



In hydraulic and civil engineering, mud is considered as a kind of viscoplastic material. The most important characteristic of
viscoplastic materials is their yield stress, which defines the point when flow occurs. Numerically, mudflow was simulated by
Bingham model (Schamber and MacArthur, 1985; Liu and Mei, 1989; Liu et al., 2016), and Herschel–Bulkley model (Bates
and Ancey, 2017; Huang and García, 1998). Both of Bingham model and Herschel–Bulkley models' ideal are discontinuous
(Mitsoulis and Tsamopoulos, 2017). Analytical solution of failure's flow of mining tailings dams was first proposed by
Jeyapalan et al., (1983). The behaviors of tailings materials were represented with Bingham model (BM). The failure of the
gypsum tailings dam in East Texas in 1966 (FGT66) was used for an application case. The study, nevertheless, just supplied
the one-dimensional profiles and the freezing time. Liu and Mei, (1989) studied a sliding flow of a high concentration mud on
an inclined plane. An analytical solution for a thin sheet of Bingham fluid was derived and verified with experimental data.
Because of the yield stress, the free surface parallels to the plane bed when the Bingham fluid is in static equilibrium. The mud
front, just like a steady gravity current, will eventually advance at a constant speed with the same profile when there is a steady
upstream discharge of mud. Huang and García, (1998) studied the spreading of a two-dimensional, unsteady mudflow on a
steep slope. The nonlinear rheological properties of the mud were described by the Herschel–Bulkley model. The von Karman
integral method was used to derive the depth-averaged continuity and momentum equations. They also discussed the influences
of shear-thinning on the free-surface profiles and spreading characteristics of the mudflow. Pastor et al., (2004) implemented
Bingham model into a depth-averaged numerical model to simulate the hyper-concentrated flows. The bottom friction was
approximated by a third-order polynomial function to save the computational time. Chen and Peng, (2006) developed a two-
dimensional two-layer model to simulate the confluence of clear water and mudflow. They used the Harten, Lax and van Leer
(1983) scheme (Harten et al., 1983) to solve the depth-averaged equations and the Strang splitting method to manage the
friction term. The model was certified by comparing the simulation results with the prediction of Pastor et al., (2004).
Given the above studies, theoretical models or two-dimensional depth-averaged numerical models can be used to simulate
mudflows in simplified conditions (Chen and Peng, 2006; Pastor et al., 2004; Li et al., 2012; Bates and Ancey, 2017). However,
the depth-integrated models are not suitable for describing the flow with strong vertical fluid particle acceleration. This
phenomenon can be seen in the case of a slope with rugged topography or mudslide overtopping a structure. Inside a complex
3D flow structure, the material might transfer from the liquid phase to the solid phase if the shear stress is less than the yield
stress. Before the material reaching this solid phase, the liquefied phase might dominate the entire flow field due to strong and
complicated shear. In other words, a model shall have the capability of describing the flow field with both solid and liquefied
phases in the computational domain simultaneously. To reach this goal, the solid phase is simulated as a fluid with extremely
high viscosity. The solid phase viscosity is large enough so that the solid deformation is much smaller than the liquefied phase
during the simulation time. However, an extremely large viscosity number indicates an extremely small time marching steps.
An implicit algorithm for the viscosity term in Navier-Stokes solver is required to overcome this issue. Navier-Stokes equations
were used to study debris flow from the decade of the 1990s. Nonetheless, most of these studies concentrated on solving 2D
problems (O'Brien et al., 1993; Assier Rzadkiewicz et al., 1997; Huang and García, 1998). Due to the development of





computers in the early twentieth century, the 3D Navier–Stokes equations have been used to study mudflow (Dai et al., 2014, Abadie et al., 2019, Wang et al., 2016). Nevertheless, the stratification effect between solid and liquid phase hasn't been invested.

This study uses full Navier-Stokes equations to describe the mud motion with PLIC VOF as a mud surface tracking algorithm. This research modifies the conventional Bi-viscosity model (CBM) by illustrating the rheology relationship between the solid and liquid phases. A yield strain rate is used to identify the plug and liquefied rheological prosperities.

The next section gives an overview of modelling approach. Section 3 presents validations of the model with analytical solutions as well as experimental data. A case study the failure of the gypsum tailings dam in East Texas in 1966 (FGT66) is provided

in Section 4. Section 5 describes a series of sensitivity analyses on the yield strain rate and grid resolution. Finally, conclusions are presented in Section 6.

## 2 Rheological Model and Numerical Algorithm

For flow rheology, Bingham model has been widely used to simulate mudflows (Coussot and Proust, 1996; Liu and Mei, 1989; Mei and Yuhi, 2001), lava flows (Griffiths, 2000) and landslides (McDougall and Hungr, 2004). The shear stress beyond the

yield stress is linearly proportional to the strain rate. If the yield stress approaches zero, the Bingham plastic fluid can be approximately treated as the Newtonian fluid. If a mud material is considered as a Bingham fluid initially at the rest, and the shear stress, $\tau$ increases, the fluid continues at rest until the shear stress reaches $\tau_0$. At that stage, the strain rate always equal to zero. Once the threshold is passed, $\tau > \tau_0$, the shear stress increases linearly with strain rate. If the external forces decrease, the strain rate decreases to zero, which occurs at $\tau = \tau_0$. This rheological model can reproduce both a static resistance to the

initiation of the flow and the stoppage of it. One of the key features of Bingham fluids is the formation of a plug zone, where velocity is constant and the strain rate is zero (Pastor et al., 2002). In the plug zone, the characteristics of solid material are presented. However, Navier-Stokes equations derived under the assumption of Eulerian fails in describing the solid motion. An alternative treatment for the plug material has to be developed. In this paper, a material with higher viscosity is adopted to present a solid phase during the simulation time. The Bingham material transfers to a material with two different viscosities,

which was originally introduced by Beverly and Tanner, (1992). However, due to the intrinsic characteristic, both Bingham model (BM) and conventional Bi-viscosity model (CBM) are not able to offer a satisfactory illustration of the stratified material. Viscoplastic models are intrinsically discontinuous at the plug/liquefied interface (Dimakopoulos et al., 2018). Therefore, the modified Bi-viscosity model (BMB) is developed. For describing the stratification feature, the plug viscosity is further increased, make the shear stress discontinuous. A yield strain rate, $\dot{\gamma}_y$ is used as the indicator to identify the corresponding

rheological prosperity. When the shear stress increases, the strain rate increases. However in cases $\dot{\gamma} < \dot{\gamma}_y$, the viscosity is defined as a value, $\mu_A$ much bigger than that in CBM, and the fluid continues at rest. Once the threshold is passed, $\dot{\gamma} \geq \dot{\gamma}_y$, the rheology returns to the Bingham liquefied fluid. This characteristic is similar to the bedload transport on a river bed. The yield


stress is analogous to the critical shear stress of an erodible bed. The plastic viscosity, $\mu_B$ represents the fluid viscosity, while the boundary viscosity, $\mu_A$ represents the viscosity of rigid material.

For Bingham model (BM),

$$\mu(\dot{\gamma}) = \begin{cases} \mu_A = \infty & ,if\ \dot{\gamma} = 0 \\ \mu_B + \dfrac{\tau_0}{\sqrt{\frac{1}{2}\dot{\gamma}:\dot{\gamma}}} & ,if\ \dot{\gamma} > 0 \end{cases} \tag{1}$$

For conventional Bi-viscosity model (CBM) (Beverly and Tanner, 1992),

$$\mu(\dot{\gamma}) = \begin{cases} \mu_A = \dfrac{\tau_0}{\dot{\gamma}_y} & ,if\ \dot{\gamma} < \dot{\gamma}_y \\ \mu_B + \dfrac{\tau_0}{\sqrt{\frac{1}{2}\dot{\gamma}:\dot{\gamma}}} & ,if\ \dot{\gamma} \geq \dot{\gamma}_y \end{cases} \tag{2}$$

For modified Bi-viscosity model (MBM),

$$\mu(\dot{\gamma}) = \begin{cases} \mu_A \gg \dfrac{\tau_0}{\dot{\gamma}_y} & ,if\ \dot{\gamma} < \dot{\gamma}_y \\ \mu_B + \dfrac{\tau_0}{\sqrt{\frac{1}{2}\dot{\gamma}:\dot{\gamma}}} & ,if\ \dot{\gamma} \geq \dot{\gamma}_y \end{cases} \tag{3}$$

where $\dot{\gamma} = \dfrac{\partial \dot{u}_i}{\partial x_j} + \dfrac{\partial \dot{u}_j}{\partial x_i}$

in which the yield stress $\tau_0$ and the viscosity $\mu_B$ are material constants that do not depend on the shear stress nor the strain rate. To describe the 3D mudflow motion, the Splash3D is adopted. This model solves three-dimensional incompressible flow with Navier-Stokes equations. The free-surface is tracked by the Volume-of-Fluid (VOF) method. The domain is discretized by the

finite volume method (FVM). The detailed explanation is provided in the text of Wu, (2004); Wu and Liu, (2009).

The model is expressed in terms of two conservation equations of mass and momentum. The mass conservation equation:

$$\frac{\partial \rho}{\partial t} + \nabla \cdot (\rho \boldsymbol{u}) = 0 \tag{4}$$

where density, $\rho$ is defined in terms of the ratio of the volume fraction of the sediments $r$ in each cell, $\rho = r\rho_s + (1-r)\rho_a$, where $\rho_s$ and $\rho_a$ are the densities of sediment and air, respectively. The volume fractions $r$ are bounded by $0 \leq r \leq 1$, where

$r = 1$: the cell fully occupies by the sediment; $0 < r < 1$: the cell includes sediment and air; $r = 0$: the cell doesn't occupy by the sediment.

The momentum conservation equation:

$$\frac{\partial \rho \boldsymbol{u}}{\partial t} + \nabla \cdot \rho(\boldsymbol{u}\boldsymbol{u}) = -\nabla p + \nabla \cdot \tau + \rho \boldsymbol{g} \tag{5}$$

where $\tau$ is viscous shear stress, calculated by the equation:

$\tau = \tau_0 + \mu(\dot{\gamma})\dot{\gamma}$

The projection method is used to solve Navier-Stokes equations. In the projection method, the momentum Eq. (5) can be described by two fractional steps:

$$\frac{\rho^{n+1}\boldsymbol{u}^* - \rho^n \boldsymbol{u}^n}{\Delta t} = -\nabla \cdot (\rho \boldsymbol{u}\boldsymbol{u})^n + \nabla \cdot (\mu^{n+1}(\nabla \boldsymbol{u} + \nabla^T \boldsymbol{u})^*) - \nabla p^n \tag{6}$$




$$\frac{\rho^{n+1}\boldsymbol{u}^{n+1} - \rho^{n+1}\boldsymbol{u}^*}{\Delta t} = -\nabla \delta p^{n+1} + \rho^{n+1}\boldsymbol{g} \tag{7}$$

where $\delta p^{n+1} = p^{n+1} - p^n$. Equation (6) is an explicit expression for the interim velocity $u^*$, referred to as the predictor step. In Eq. (6), all forces except for gravity and pressure gradient are included. Equation (7) is termed the projection step. Combining Eq. (6) and (7) exactly produces the time discretization of Eq. (5):

$$\frac{\rho^{n+1}\boldsymbol{u}^{n+1} - \rho^n \boldsymbol{u}^n}{\Delta t} = -\nabla \cdot (\rho \boldsymbol{uu})^n + \nabla \cdot (\mu^{n+1}(\nabla \boldsymbol{u} + \nabla^T \boldsymbol{u})^*) - \nabla p^{n+1} + \rho^{n+1}\boldsymbol{g} \tag{8}$$

In this study, the fluids are assumed to be Non-Newtonian fluids. Viscous forces are incorporated into the predictor step to

estimate the cell-centered velocity field by using the * time step velocity field. This implicit approximation eliminates the limitation of the time step for the stability criteria.

The net viscous stress on the control volume is calculated by applying the divergence theorem to the volume integral of the local stress. This is determined by totalizing of the dot product of the face normal vector with the local velocity gradient multiplied by the face area

$$\nabla \cdot (\mu^{n+1}(\nabla \boldsymbol{u} + \nabla^T \boldsymbol{u})^*) = \sum_f \mu_f^{n+1} A_f \left[\widehat{\boldsymbol{n}}_f \cdot (\nabla \boldsymbol{u}_f + \nabla^T \boldsymbol{u}_f)\right]^* \tag{9}$$

where $\mu_f$, $A_f$, $\widehat{\boldsymbol{n}}_f$, and $u_f$ is viscosity, area, normal vector, and velocity of the cell face, respectively.

There are two stability conditions of $dt$ need to satisfy when solving Navier-Stokes equations:

$$dt_c < C_r \frac{dl}{Max(|\boldsymbol{u}|)} \tag{10}$$

$$dt_\mu < V_\mu \frac{(dl)^2}{Max(\mu_{eff})} \tag{11}$$

Where $dt_c$ is the time step restricted by the advection term, $C_r$ is Courant number, which is defined as $C_r = Max(|\boldsymbol{u}|)dt/dl$, $dl$ is the measure of the cell size, $dt_\mu$ is the time step restricted by the diffusion term, $V_\mu$ is the viscous number which is defined as $V_\mu = Max(\mu_{eff})dt/(dl)^2$.

The stability criteria of the viscous term in Navier-Stokes equations will be introduced if a large viscosity parameter is imposed. However, this small time-step restriction can be relaxed by adopting the implicit scheme. The viscous implicitness $\theta$ is used

to calculate the velocity $\boldsymbol{u}$ at time level $\theta$, and $\boldsymbol{u}^\theta = (1-\theta)\boldsymbol{u}^n + \theta \boldsymbol{u}^{n+1} + 1$. In this study, $\theta$ is given as unity which implies a fully implicit treatment, and $dt_\mu$ is no longer restricted by Eq. (11).

## 3 Validation

To demonstrate the accuracy of the model, two cases of Bingham fluid are simulated. The yield strain rate, $\dot{\gamma}_y = 0.0$ s$^{-1}$ is used to simulate. The results are compared with analytical solutions and laboratory experiment data.




## 3.1 Bingham fluid driven by pressure


Byron-Bird et al., (1983) provided analytical solutions for the Bingham flow in a channel, driven by a pressure gradient $P_0 - P_L$, illustrated in Table 1. The channel was depicted as the length $L$ and the width $2B$. The no-slip boundary condition was applied on the surfaces of the channel (Fig 1).

The "yield surface" is located at $x = x_0$ where $x_0 = \frac{\tau_0 L}{P_0 - P_L}$. The velocity in the "plug region" $v_z^>$ and the "liquefied region" $v_z^<$

is defined:

$$v_z^> = \frac{(P_0 - P_L)B^2}{2\mu_B L}\left[1 - \left(\frac{x_0}{B}\right)^2\right] - \frac{\tau_0 B}{\mu_B}\left[1 - \frac{x_0}{B}\right] \qquad (-x_0 \le x \le x_0) \tag{12}$$

$$v_z^< = \frac{(P_0 - P_L)B^2}{2\mu_B L}\left[1 - \left(\frac{x}{B}\right)^2\right] - \frac{\tau_0 B}{\mu_B}\left[1 - \frac{x}{B}\right] \qquad (x_0 < x \le B \text{ and } -B \le x < -x_0) \tag{13}$$

$0 \le x_0 \le B$. Therefore, $\tau_0 = \frac{x_0(P_0 - P_L)}{L}$ is further limited at the region of:

$$0.0 \le \tau_0 \le \frac{B(P_0 - P_L)}{L}$$

When $x_0 = 0.0$, or $\tau_0 = 0.0$, the fluid is Newtonian fluid.

Four cases are validated with Eq. (12)-(13), including a Newtonian case and three Bingham cases with different parameters such as channel length $L$, channel width $2B$, one end's pressure $P_0$, Bingham viscosity $\mu_B$, and yield stress $\tau_0$, which are shown in Table 1. Fig 2 shows good agreements between theory and numerical results for all runs. The most important feature of a Bingham fluid is the plug zone (Fig 2 (b)-(d)), which cannot be seen in the Newtonian fluid (Fig 2 (a)). Notice that the velocity

of a Bingham fluid is constant in the plug region. In this region, the rate of change of velocity (strain rate) is equal to zero. In the liquefied region, the strain rate is greater than zero and the stress-strain relation of the fluid is dependent on the plastic viscosity $\mu_B$. These figures demonstrate that the present numerical model can be used to simulate the rheological behavior of Bingham fluids.

Table 1. Summary of validation of flow of Bingham fluid in a channel.

| Case | B (m) | L (m) | $P_0$ (N) | $P_L$ (N) | $\mu_B$ (Pa s) | $\tau_0$ (Pa) | Cell size $dx/B$ |
|---|---|---|---|---|---|---|---|
| 1 | 1 | 5 | 4 | 0 | 0.6 | 0.0 | 0.025 |
| 2 | 1 | 5 | 4 | 0 | 0.6 | 0.24 | 0.025 |
| 3 | 2 | 12 | 20 | 0 | 3. | 1.7 | 0.025 |
| 4 | 5 | 20 | 15 | 0 | 12.0 | 2.4 | 0.025 |






## 3.2 Spreading of Bingham fluid on an inclined plane

The validation of the spreading of Bingham fluid on an inclined plane is set up based on the experiment of Liu and Mei, (1989). In their publication, they didn't only provide laboratory data but also provide an analytical solution of the spreading of Bingham fluid on an inclined plane presented in Fig 4 (a). Kaolinite was mixed with tap water to simulate mud. The mud was contained

in a reservoir. When the adjustable gate was opened revealing a gap $H = 0.0051$ m, the mud flowed down on an inclined dry bed with the inclined angle $\theta = 0.9°$. The fluid density $\rho = 1106$ kg m$^{-3}$, the yield stress, $\tau_0 = 0.875$ Pa, the viscosity of the plug zone, $\mu_A = 1. e10$ Pa s, the viscosity of liquefied zone, $\mu_B = 0.034$ Pa s. A two-dimensional domain is set up as 3.5 x 0.2 m (Fig 3), is discretized into a regular mesh with grid size dx = 2.3 mm, dz = 2.0 mm. Fig 4 (a) presents that the numerical result from Bingham model validates well with the theoretical solution as well as experimental data from Liu and Mei, (1989).

Fig 4 (b) shows the spreading of mud on an inclined plane versus time. The mudflow develops a self-similar front when time t > 8.0 s. Because of the yield stress, the free surface parallels to the plane bed when the Bingham fluid is in static equilibrium. The mud front, just like a steady gravity current, eventually advances at a constant speed with the same profile when there is a steady upstream discharge of mud.

## 4 Case study - The failure of the gypsum tailings dam in East Texas in 1966 (FGT66)

### 4.1 Numerical setup

The present numerical model is applied to simulate the failure of the gypsum tailings dam in East Texas in 1966 (FGT66), shown in Fig 7 (b). The impoundment was rectangular, and had reached a height of 11 m by the time failure took place. The slide was caused by seepage at the toe of the slope, and affected a length of 140 m of the dike, extending 110 m into the impoundment lagoon. An estimated 80,000 m$^3$ - 130,000 m$^3$ of gypsum were released in this flow failure. The released material

travelled 300 m before stopping, with an average velocity of 2.5 – 5.0 m/s (Jeyapalan et al., 1983). The numerical setup is based on the geometry reported in Jeyapalan et al., (1983), shown in Fig 5. The size of the impoundment is 280 x 110 x 11 m, and the breach is 120 m in width, 20 m in thickness, and the center cross-section of the breach is located at y = 220 m. The computational domain (510 m in length, 400 m in width, and 12 m in height) is discretized into a uniform mesh with a grid size dx = 2.0 m, dy = 2.0 m, dz = 1.0 m. The number of the total grid is 612,000. The bottom boundary (at z = 0 m) is no-slip

boundary condition, the downstream (x = 400 m) and lateral boundaries (y = 0 m and y = 400 m) are free-slip boundary conditions. The material of the gypsum tailings is assumed as mud. Based on the values suggested by Jeyapalan et al., (1983), the yield stress of the mud is $\tau_0 = 10^3$ Pa, the viscosity of the liquefied zone is $\mu_B = 50$ Pa s, and the density is $\rho = 1400$ kg m$^{-3}$. The viscosity of the plug zone is $\mu_A = 1. e10$ Pa s.





**4.2 The results from three different rheological models**

In earlier publications such as Jeyapalan et al., (1983); Pastor et al., (2002), Bingham plastic rheological model was used to represent the behavior of tailings materials during failure flow. In this study, Splash3D is coupled with three different rheological models to reproduce the FTG66, shown in Fig 6. The first rheological model is Bingham model (BM) ($\dot{\gamma}_y = 0.0$ s$^{-1}$). The simulations result is shown in Fig 6 (a). The mud thickness reduces gradually from the breach to the downstream toe. A sliding mud body is thicker at the breach center area. BM can simulate the stoppage of the material at t ≥ 90 s. The second

rheological model is a conventional Bi-viscosity model (CBM). The result shown in Fig 6 (b)-(c) display the gypsum tailings produced by CBM with $\dot{\gamma}_y = 1 \times 10^{-4}$ s$^{-1}$, and $\dot{\gamma}_y = 2 \times 10^{-1}$ s$^{-1}$, respectively. In the CBM, the high-viscosity fluid is used for representing the solid phase and the low-viscosity fluid is used for representing the liquefied phase. The tailing shapes are very similar to each other. However, the mud profiles are different. The 'ridge' at the center of the breach (Fig 6 (b)) is replaced by a smooth hump (Fig 6 (c)). It is due to the differences between $\mu_A$ and $\mu_B$ of two cases. All of the cases shown in this section

use the same values of viscosity $\mu_B$ for the liquefied zone, and yield shear stress $\tau_0$. The viscosity for the plug zone $\mu_A$ is depend on the yield strain rate $\dot{\gamma}_y$, $\mu_A = \tau_0 / \dot{\gamma}_y$. The bigger $\dot{\gamma}_y$, the smaller $\mu_A$, the closer between $\mu_A$ and $\mu_B$. The rheological relationships for the yield strain rate varying from  $\dot{\gamma}_y = 1 \times 10^{-4}$ s$^{-1}$ to $\dot{\gamma}_y = 2 \times 10^{-1}$ s$^{-1}$ are shown in Fig 6 (d). The horizontal axis for the strain rate is exaggerated to show the differences of each  $\dot{\gamma}_y$.

In Fig 6 (a)-(c), we observe that CBM can reproduce the result from the BM as long as the yield strain rate is small enough. In

the following discussion, the results from CBM are not shown because they are nearly identical to the result of BM. However, both BM and CBM are not able to offer a satisfactory illustration of the final shape as shown in the photo (Fig 7 (b)). One possible reason is the stratification phenomenon developed naturally under the effect of gravity. The stratification effect is similar to a tamp effect which will strengthen the material. However, when the material liquefies, the material property goes back to the Bingham liquefied material. To describe this phenomenon, the plug zone viscosity $\mu_A$ is defined as much larger

than $\tau_0 / \dot{\gamma}_y$ (According to Eq. (3)). The mud tailings given by MBM is shown in Fig 6 (e). Compared to the aerial photograph shown in Fig 7 (b), we can see that the result from the present model at the freezing time t = 110 s has an outlook much closer to the aerial photo (Jeyapalan et al., 1983).

Fig 8 and Fig 9 shows the time evolution of surface velocity of the mudflow from t = 0 ~ 110 s using BM and MBM, respectively. The velocity at the initial stage (about t = 0 ~10 s) of BM is higher than that of MBM. However, the simulated

mudflow in BM stops earlier (around 70 – 90 s). The surface velocity approaches zero from t = 70 s and the flows stop totally at t = 90 s. The gypsum tailings distance is around 220 m, and the mean velocity is around 2.4 – 3.1 m s$^{-1}$ shown in Fig 8. On the other hand, the simulated mudflow by using MBM can go further because the yield strain rate limits the velocity at the initial stage (Fig 9). The mud starts to liquefy and collapse in a small region near the breach in the first 10 s. The spreading shape of the tailing is symmetric along the centerline of the breach during t = 0 – 20 s. Because of the supply of the tailing

from the impoundment is asymmetric, the spreading shape gradually became asymmetric when t > 20 s. The maximum velocity





of the released tailings occurs when t = 30 s. The flow velocity gradually decreases. After t = 90 s, the strain rate is smaller than the yield strain rate and the tailings gradually stop moving at t = 110 s. The Gypsum tailings distance is around 310 m, and the mean velocity is around 2.8 – 3.4 m s$^{-1}$ shown in Fig 9.

The predictions of inundation distance, freezing time, and mean velocity by observed values, theoretical data, and numerical model results of Jeyapalan et al., (1983); Pastor et al., (2004); and Chen and Peng, (2006) are listed in Table 2 for comparison. The simulation results of the MBM are very close to the observed values, and the accuracy is better than the two-dimensional models of Jeyapalan et al., (1983); Pastor et al., (2004); and Chen and Peng, (2006). Not only freezing time, and mean velocity but also the value of inundation distance receives good agreements with observed values.

Table 2. The summary of the observed values, the historical results and the result using Splash3D model coupled BM and
MBM.

|  | Inundation distance (m) | Freezing time (s) | Mean velocity (m s$^{-1}$) |
|---|---|---|---|
| Observed values (Jeyapalan et al., 1983) | 300 | 60 ~ 120 | 2.5 ~ 5.0 |
| Theoretical results from charts (Jeyapalan et al., 1983) | 550 | 132 | 4.2 |
| Jeyapalan et al., (1983) | 670 | 116 | 6.0 |
| Pastor et al., (2004) | 330 | 120 | 2.75 |
| Chen and Peng, (2006) | 360 | 120 | 3.0 |
| Bingham model (Fig 8) | 220 | 70 – 90 | 2.4 – 3.1 |
| Modified Bi-viscosity model (Fig 9) | 310 | 90 – 110 | 2.8 – 3.4 |

Fig 10 reveals the MBM result of strain rate and the plug/liquefied zones profiles on the center cross-section of the breach (y = 220 m). The vertical axis is ten times exaggerated. The yield strain rate $\dot{\gamma}_y = 0.2$ s$^{-1}$ is chosen in the simulation. The color bar is set from 0.0 to 1.0 s$^{-1}$ emphasizes the interface between the plug zone and liquefied zone at $\dot{\gamma}_y = 0.2$ s$^{-1}$. The liquefied
zone is located at the breach's front when t = 10 s, and gradually shifts to the region near the ground after t = 10 s. It is due to the large shear stress near the bottom. However, another part of the material keeps in the shape of solid. During the period of 10 – 40 s, the discontinuity between solid and liquefied zone is presented. At t = 90 - 110 s, the liquefied zone shrinks gradually and disappears at t = 110 s due to the zero velocity of the entire flow field. This zero-velocity phenomenon is very close to the real landslide situation with the velocity ceases to zero eventually.

**4.3 Difference between the BM and MBM**

Fig 11 shows the velocity magnitude of mudflow in BM as well as MBM on the center cross-section (y = 220 m). For BM, the entire fluid material tends to slide down and moves faster than the results in MBM. The maximum moving velocity of the





tailing front at t = 10 s is approximately 6.0 – 8.0 m s$^{-1}$ and decreases sharply during t = 10 – 40 s. It makes the inundation distance (around 220 m) at t = 110 s shorter than the one in MBM. In the MBM results, the maximum moving velocity is about

5.0 – 7.0 m s$^{-1}$ which is slightly smaller than the results in BM. However, the MBM result takes a longer time to reach the zero-velocity stage and the resulting inundation distance is longer than the BM result. The inundation distance (310 m) predicted by MBM is very close to the field observed (300 m).

Fig 12 shows that MBM can clearly present the discontinuous interface between the plug and liquefied zones in the strain rate profiles. However, this interface can't be seen in BM. The sliding slope of the mudslide is developed automatically by MBM.

This is big progress in studying mudslide flows.

## 5 Discussion

### 5.1 Resolution sensitivity analysis

The sensitivity tests on the grid resolution are performed in this section. Three cases of BM and nine cases of MBM are performed with the resolutions varying from dx: 1.8 – 2.2 m, dy: 1.8 – 2.2 m, and dz: 0.8 – 1.2 m. Fig 13 shows the final

profiles simulated by BM and MBM. The results show that BM is less sensitive to the resolution compared to MBM. The results from MBM present that dz is more sensitive than dx, and dy. The inundation distance is shorter for dz = 1.2 m (blue lines) and longer for dz = 0.8 m (red lines). To give an overview, the inundation width and inundation distance are convergent as dz < 1.2 m.

### 5.2 Sensitivity analysis of the yield strain rate

In MBM, the yield strain rate $\dot{\gamma}_y$ defines the fluid behavior in the regime of either plug zone or liquefied zone. If the yield strain rate is zero, the material returns to Bingham fluid which is not able to describe the stratification effect. The mud spreads like a circle around the breach. As the yield strain rate becomes higher, the plug zone is larger, and the fluid is harder to transfer from solid to liquid. The behavior of the front position is not monotone when $\dot{\gamma}_y$ increases. There are two behaviors of the mud front, presented in Fig 14. In terms of $0.0 \leq \dot{\gamma}_y \leq 0.3$ s$^{-1}$, when $\dot{\gamma}_y$ increases, the inundation widths are narrower; however, the

inundation lengths are longer. That means the mud tends to flow towards downstream direction strongly instead of spreading both sides. In terms of $0.3 \leq \dot{\gamma}_y \leq 0.6$ s$^{-1}$, when $\dot{\gamma}_y$ increases, the inundation widths are narrower and the inundation lengths are shorter. That means the mud spreading is limited in both downstream directions and both sides' directions.

Rheological properties of hyper-concentration are generally formulated as a function of the concentration of the fluid material. Julien, (2010) recommended empirical formulas with the exponential relationships for yield stress and viscosity at large

concentrations of fines. The typical values of coefficients for different types of muds, clays, and lahars are presented in Table 3. Kaolinite and Typical soils are utilized to describe the features of BM and MBM in this section. Eighteen numerical cases including nine Bingham cases and nine modified Bi-viscosity cases with different concentration $C_v$ are performed. The yield


shear stress and the viscosity are presented in Table 4. The yield strain rate is specified as $\dot{\gamma}_y = 0.0$ s$^{-1}$ for the Bingham cases and $\dot{\gamma}_y = 0.2$ s$^{-1}$ for the modified Bi-viscosity cases. The goal of these tests is to find a material in which the fluid property is

similar to the tailing material in FGT66. A similar flooding profile is a key. Fig 15 shows the simulation results with different concentration $C_v$ by BM and MBM. The final shape of Kaolinite at $C_v = 0.5$ (red line in Fig 15 (b)) has the best fit to the final shape of numerical tailing dam failure (red line in Fig 14) due to the similarity $\tau_0$ between two materials: Kaolinite $\tau_0 = 1580$ Pa and the gypsum tailings material $\tau_0 = 1500$ Pa. Among the results, MBM present more realistic results than BM in term of the irregular boundary curvatures. The yield strain rate, $\dot{\gamma}_y = 0.2$ s$^{-1}$ can be used to simulate not only the gypsum tailings

material but also many kinds of mud with different  sediment concentration $C_v$.

Table 3. Coefficients a, b, c of the yield strength (yield shear stress) and viscosity relationships Julien, 2010

| Material | Liquid limit $C_v$ | Yield strength in Pa $\tau_0 = a10^{bC_v}$ | | Viscosity in Pa s $\mu_B = 0.001 \times 10^{cC_v}$ |
|---|---|---|---|---|
|  |  | a | b | c |
| Bentonite | 0.05 – 0.2 | 0.002 | 100 | 100 |
| Sensitive clays | 0.35 – 0.6 | 0.3 | 10 | 5 |
| Kaolinite | 0.4 – 0.5 | 0.05 | 9 | 8 |
| Typical soils | 0.65 – 0.8 | 0.005 | 7.5 | 8 |
| Granular material | - | - | 2 | 3 |

. Table 4. The yield shear stress and viscosity of Kaolinite and Typical soils

|  | Kaolinite | | | | | Typical soils | | | |
|---|---|---|---|---|---|---|---|---|---|
| $C_v$ | 0.42 | 0.44 | 0.46 | 0.48 | 0.5 | 0.65 | 0.7 | 0.75 | 0.8 |
| Yield shear stress | 301.28 | 456.00 | 690.19 | 1.04e3 | 1.58e3 | 374.95 | 889.14 | 2.11e3 | 5000.00 |
| Viscosity | 2.29 | 3.31 | 4.79 | 6.92 | 10.00 | 158.49 | 398.11 | 1000.00 | 2.51e3 |

## 6 Conclusion

The goal of this study is to provide a modification of conventional Bi-viscosity model (CBM) to describe the mudslide with the stratification effect. In the plug zone, the solid behavior is described by a fluid material with large viscosity. This model is then integrated into Splash3D model, which is resolving full Navier-Stokes equations using PLIC VOF as a mud surface tracking algorithm. In this paper, validations are made on the pressure channel flow and mudflow on an inclined plane. The validation results show that Splash3D with rheological msodels can simulate the flow motions and have good fits to the

analytical solution and laboratory data.



Both Bingham model (BM) and conventional Bi-viscosity model (CBM) are then used to simulate FGT66. From the simulated results, one can notice that with a small yield strain rate $\dot{\gamma}_y$, results from BM and CBM are nearly identical. However, both results cannot reproduce the appearance of the mud flood. To introduce the stratification effect of the tailings impoundment, a modification is raised to simulate the FGT66. A series of plug viscosity is applied with sensitivity analysis. The results show

that MBM can present the detail irregular boundary of the flood appearance. Both predicted flood distance and flood speed are very close to the field data. The MBM illustrates the process that the plug zone and liquefied zone develops. The simulations show the initiation of the mudslide, and then the development of the slip surface, the flooding process, and velocity ceasing process. One shall note that the slip surface is developed automatically without empirical equations. By comparing the results of BM, MBM, and field data, one can conclude that the liquefied tailings are under the effect of stratification, and the

stratification effect is presented in the increased plug viscosity in the MBM.

Finally, MBM with the feature of discontinuous shear stress can simulate the tamping and stratification effects of the sliding material. The MBM can describe all the sliding processes including the development of the slip surface. Compared to the conventional BM, MBM can provide more details on the material in nature.

## Acknowledgements


The research leading to these results has received funding from the Ministry of Science and Technology (MOST), Taiwan, under the project number: 107-2116-M-008-012 and the Centre of Excellence for Ocean Engineering, National Taiwan Ocean University.

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

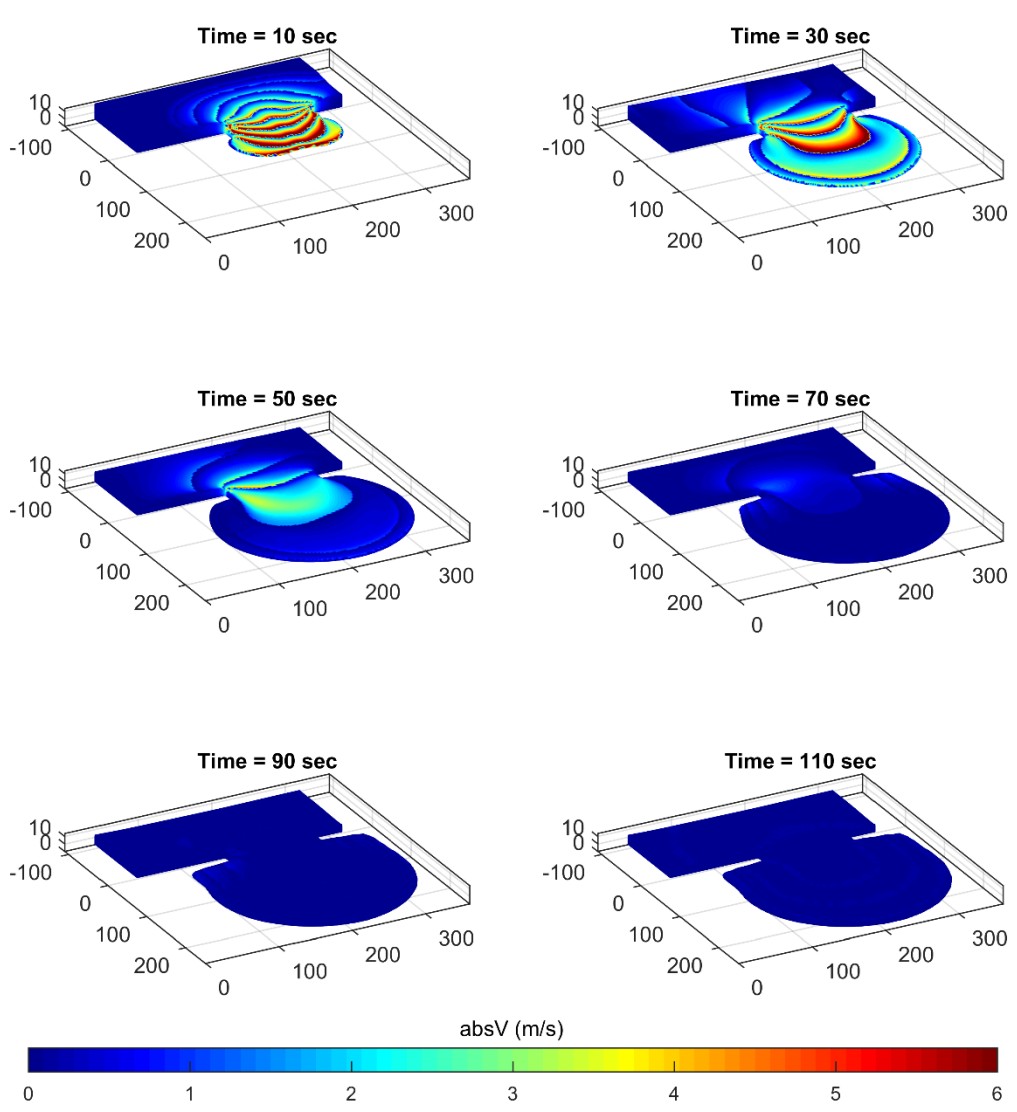

**Fig 8. Snapshots of surface velocity simulated by BM.**




**Fig 9. Snapshots of surface velocity simulated by MBM.**



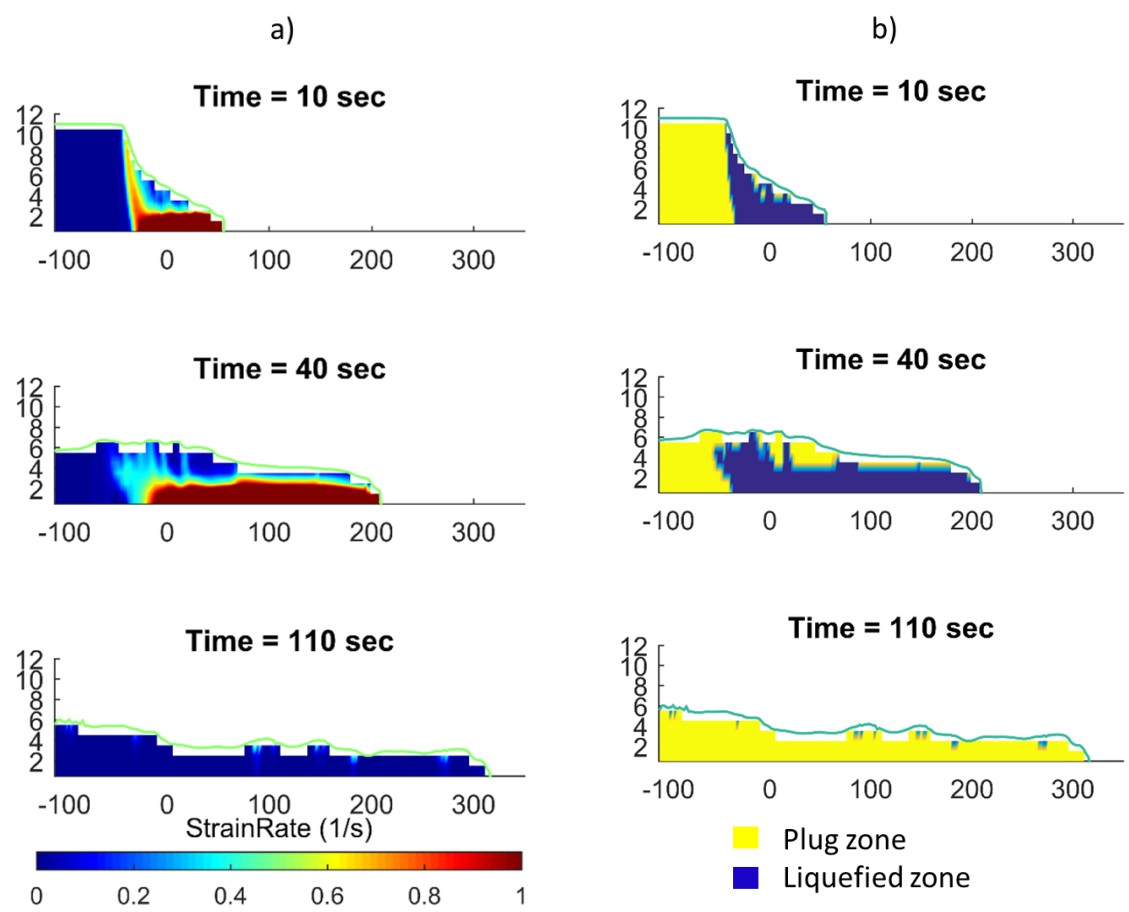


**Fig 10.** **Snapshots of a) Strain rate value, the color bar is set from 0.0 to 1.0 s⁻¹ to emphasizes the interface between the plug zone and liquefied zone at $\dot{\gamma}_y = 0.2$ s⁻¹. b) The separation of plug zone and liquefied zone on the center plane of the breach (y = 220 m).**




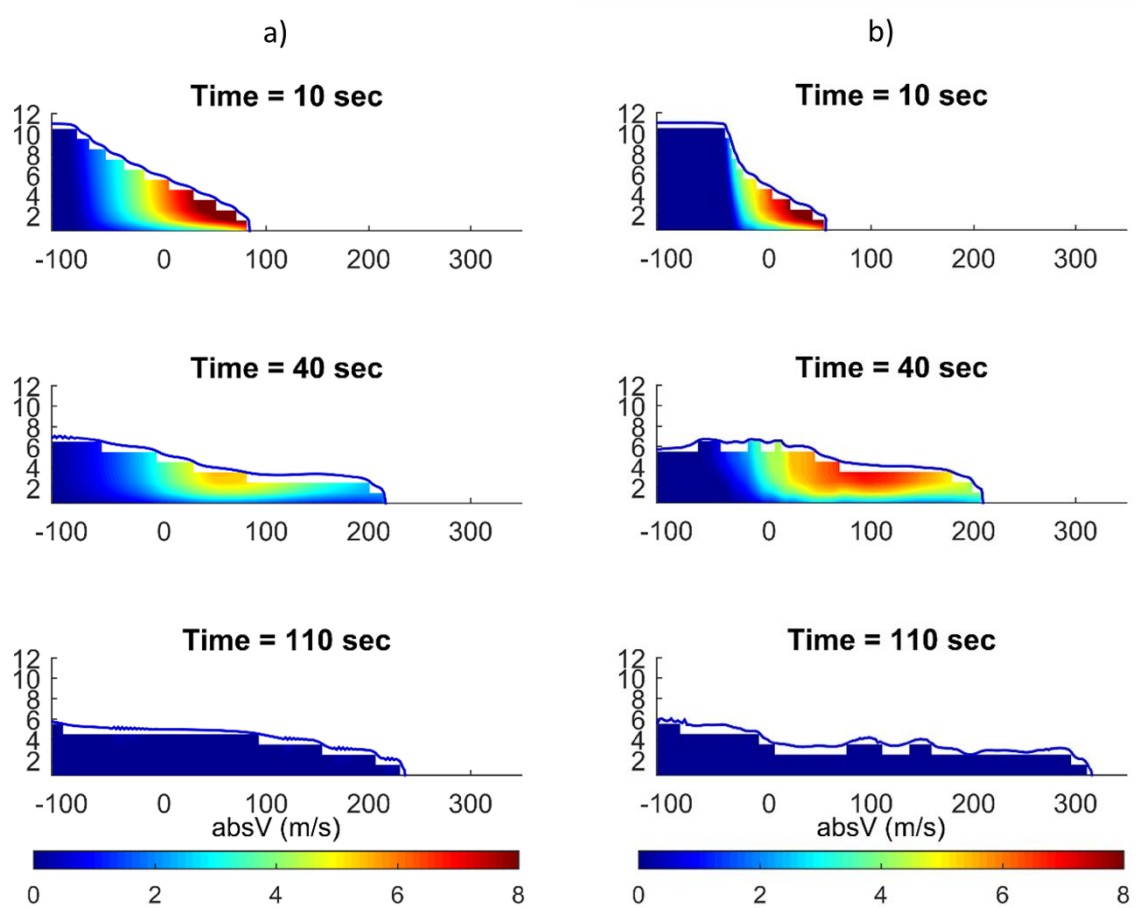

**Fig 11.** Snapshots of velocity magnitude simulated by a) Bingham model (BM) b) modified Bi-viscosity model (MBM) on the center plane of the breach (y = 220 m).

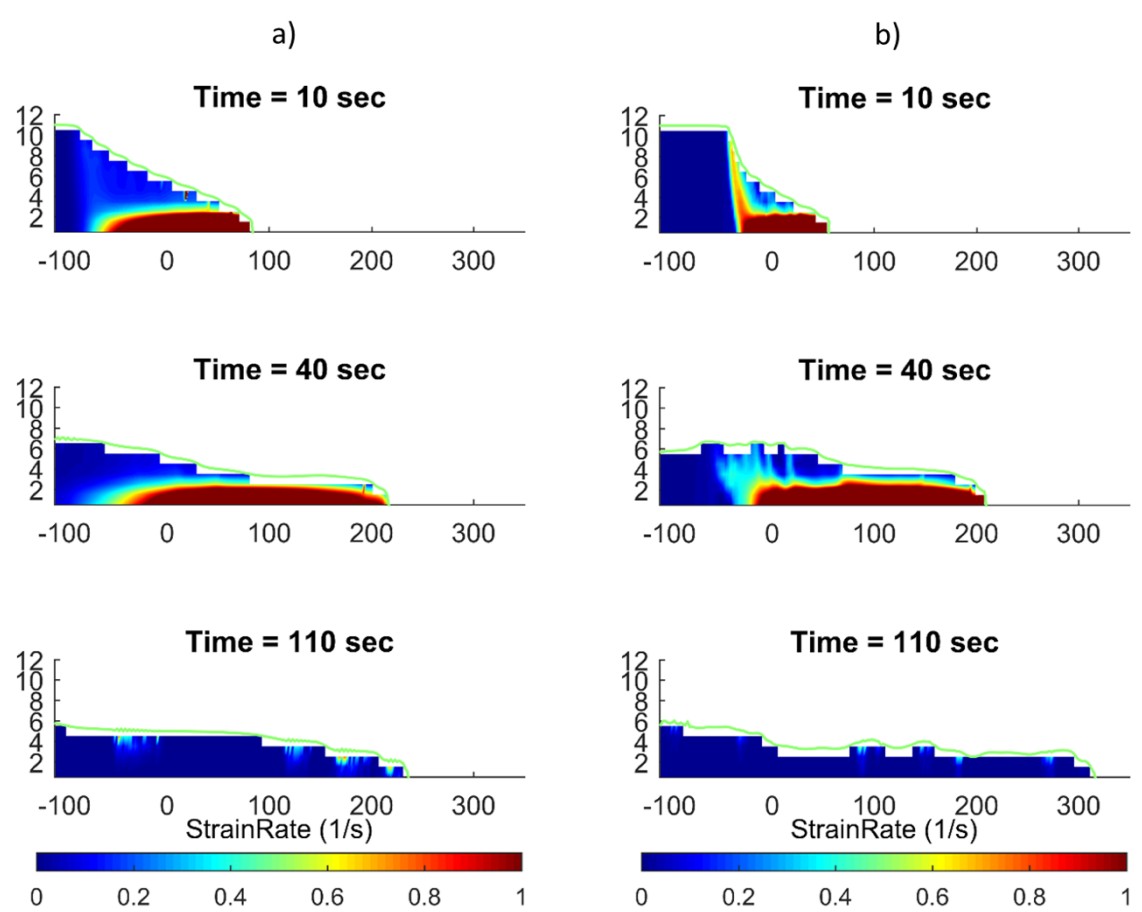

**Fig 12.** **Snapshots of Strain rate simulated by a) Bingham model b) modified Bi-viscosity model on the center plane of the breach (y = 220 m).**


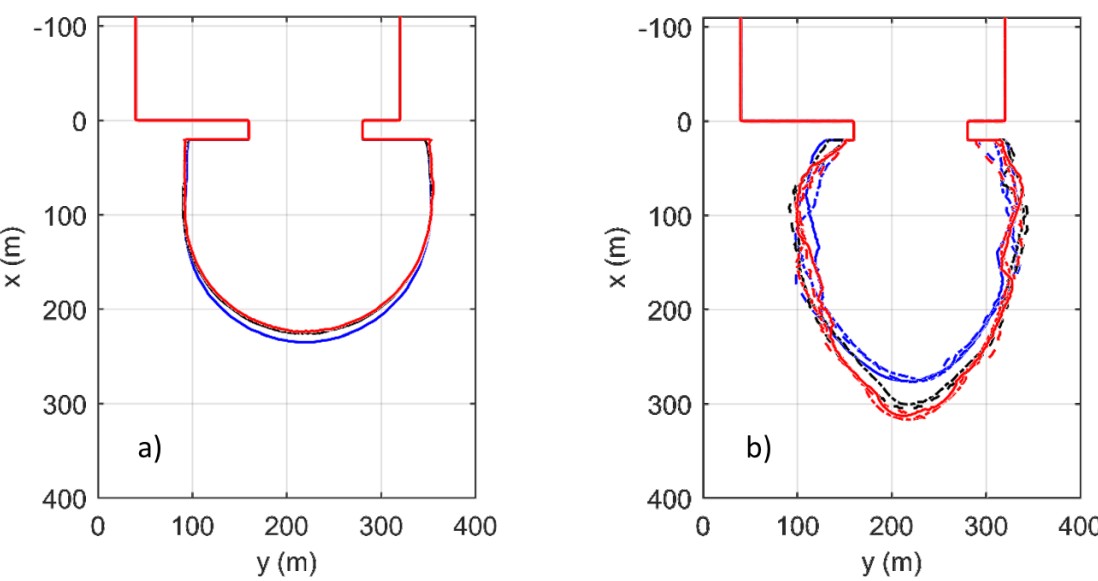

**Fig 13.** The mesh sensitivity of the liquefied tailings in a) Bingham model b) modified Bi-viscosity model. Red lines: dz = 0.8, black lines: dz = 1.0, blue lines: dz = 1.2, solid lines: dx = dy = 1.8, dashed lines: dx = dy = 2.0, dash-dot lines: dx = dy = 2.2.

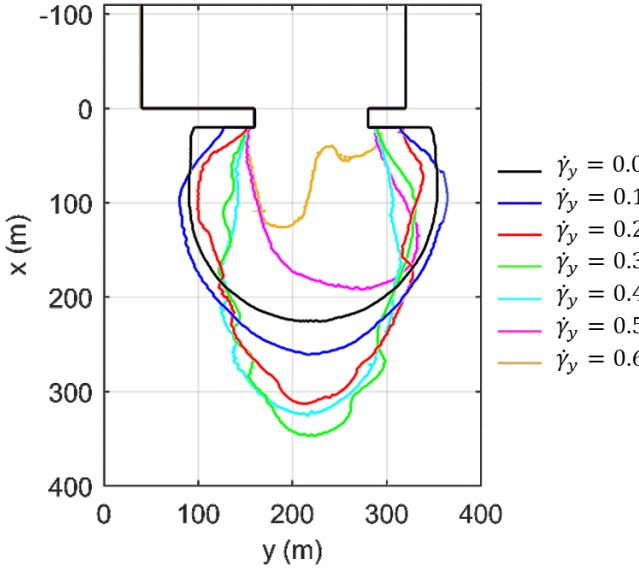

**Fig 14.** The shape of the liquefied tailings is depended on the yield strain rate (unit: s$^{-1}$)




**Fig 15.** **a-b) The liquefied tailings of Kaolinite provided by BM and MBM, respectively; c-d) The liquefied tailings of Typical soils provided by BM and MBM, respectively.**