# Peer review of "Model development for simulating mudslide and the case study of the failure of the gypsum tailings dam in East Texas in 1966"

_Natural Hazards and Earth System Sciences, 2020_

## Referee Comment (RC1) · Anonymous Referee #1 · 22 Jun 2020

General Comments:

In this paper, the authors use the Navier-Stokes-type equations to describe the mud motion with a PLIC VOF code to track the mud surface. This is primarily achieved by modifing the conventional Bi-viscosity model by incorporating the rheology relationship between the solid-type and liquid-type state of the material. A yield strain rate is used to identify the plug and liquefied regime. Some validations of the model are presented with analytical solutions and laboratory experimental data. Furthermore, a case study of the failure of the gypsum tailings dam in East Texas in 1966 is provided. And, a series of sensitivity analyses on the yield strain rate and grid resolution are

presented. From numerical/simulation point of view the results are interesting as they capture some basic observed phenomena, the audience of NHESS might find it useful. However, the physical and mechanical aspects of the manuscript are weak, and could be substantially improved. Writing is often very strange and less scientific/professional. I would suggest to take help from a professional English editor. Referencing of relevant, recent literature could be improved.

Title: I don't see much of the model development, rather simulations. So, I would suggest to remove "Model development for" from the Title. If not, please justify why you want to keep it.

Abstract: Abstract needs to be substantially improved/re-written, here are some examples.

"Mudslides, avalanches, and mine dam-breaks can be serious disasters and cause severe damages but the detailed flow field description has not been completed yet.": This is not true, at least for avalanche, mud and debris flows, see, e.g., Mergili et al., 2020 (https://doi.org/10.5194/nhess-20-505-2020; https://doi.org/10.5194/hess-24-93-2020); Yu et al., 2020 (https://doi.org/10.5194/nhess-20-727-2020). There are too may acronyms in the abstract, making it difficult to follow/remember. "a yield strain rate is used to identify the plug and liquefied rheological prosperities. The viscosity term is solved by implicit iteration.": Do you need to say these in the abstract? "The slip surface is developed automatically without empirical equations.": What does it mean? "By comparing the results of BM, CBM, and MBM to the field data, we conclude that the liquefied tailings are under the effect of stratification, and the stratification effect is presented in the extremely high plug viscosity in the Splash3D model.": Isn't it meaningless as far as you are talking about the superiority of MBM?

Specific Comments/Technical Improvements:

L23: Provide References, also in the following text whenever necessary.

L26-28: "the information is not complete": what do you want to say, not clear. "one-dimensional profiles": of what? "contradictory": with what? Please make these aspects clear.

L32: –> (means you may consider to include, or change to): or the Coulomb-viscoplastic model (Pudasaini and Mergili, 2019: https://doi.org/10.1029/2019JF005204).

"Both of Bingham model and Herschel–Bulkley models' ideal are discontinuous": please check English. I would suggest to take help from a professional English editor.

L46: –> Recently Pudasaini and Mergili (2019) proposed a first-ever multi-mechanical, multi-phase mass flows model that employs pressure- and rate-dependent Coulomb-viscoplastic rheology that is very flexible to be applied to the wide range of geophysical mass flows.

L51: "with strong vertical fluid particle acceleration." –> "with strong vertical fluid particle acceleration (Domnik et al., 2013: https://doi.org/10.1016/j.jnnfm.2012.03.001, https://doi.org/10.1016/j.jnnfm.2013.07.005)."

L54: "stress" –> "stress (von Boetticher et al., 2016, 2017: http://dx.doi.org/10.5194/gmd-9-2909-2016; https://www.geosci-model-dev.net/10/3963/2017/; Domnik et al., 2013; Khattri and Pudasaini, 2019: https://doi.org/10.1016/j.matcom.2019.03.014)"

L55-60: Please improve writing.

L74: –> and multi-phase mass flow (Pudasaini and Mergili, 2019)

L74-82: Very basic, could be removed.

L82: "However, Navier-Stokes equations derived under the assumption of Eulerian fails in describing the solid motion.": What does it mean? Not clear.

L86-89: Please explain how you would obtain stratification, because this is the main aspect of this MS. And, it seems to be mostly parameter fit, increase mu_A as high as you get what you want. So, what is the mechanical aspect of this paper, and how can you justify using those mu_A values. This is a critical aspect here. Please clearly justify, discuss.

L87: –> but, this could be regularized (Papanastasiou, 1987: https://doi.org/10.1122/1.549926; Domnik et al., 2013; Pudasaini and Mergili, 2019; Yu et al., 2020).

L95-101: Can you include the effect of particle concentration in the slurry viscosity as in von Boetticher et al. (2016)? Which value of mu_A would be chosen, how/why and are there any experimental/field evidence? Please discuss. Also write explicit expression for "dot(gamma):dot(gamma)", helping the reader. Does (3) avoid discontinuity, or increase it? Consideration of (3) does not seem to constitute a new model as mentioned in the Title. This appears to be defined for simulation purpose to try to obtain pattern seen in the field event. If not, please provide arguments.

L108: It is a bit confusing. Sometime you talk about solid sometime about fluid, and now it appears that the material is dry. What type of material, and composition is considered? Further, how do you determine r? As the material density changes during motion, do you have a transport equation for r? The VOF method was also used by Yu et al. (2020) to track the interface between the tailings and air. Also, improve English in the following lines.

L116-125: (i) It is not clear if you are using (6) and (7), or only (8). (ii) If (6) and (7) are used what is the advantage? (iii) Can you provide some references? Or, do you develop all these equations (3)-(11)? The readers would wonder. (iv) Can you define what is solid and fluid? It is completely confusing. You mention sediment and fluid, but never mention what material/composition you are considering. Are you using solid + fluid two-phase mixture material? Does not seem so.

L138: "a large viscosity parameter is imposed": again, no physical explanation.

L158-159: Could be removed, not necessary, very basic.

Table 1 is not needed, all parameters are in Fig.

L176: "Because of the yield stress, the free surface parallels to the plane bed": But not in the front.

L177: "The mud front, just like a steady gravity current, eventually advances at a constant speed": Can you show it analytically, or you are talking only about numerical results?

L190: "lateral boundaries (y = 0 m and y = 400 m) are free-slip boundary": Strange! Why don't you explain why this condition is used while the base is no-slip?

L193: The value of mu_A is huge! Can you justify it mechanically, or prove with evidence as the paper is mainly based on such values?

L196: "Splash3D": Please explain it briefly, not all the readers might know it.

L211-215: The most important aspect of this paper is concerned with the generation of the stratification pattern observed in the field event. However, the authors could not make it quite clear what exactly is the mechanism in their model that produces these patterns. This is perhaps the major drawback of the MS. How can just gravity do that job? Please make it clear.

L222-223: "the simulated mudflow by using MBM can go further because the yield strain rate limits the velocity at the initial stage": this is great! Seems to be the best thing in the paper!

L232-233, 250-252: "Not only freezing time, and mean velocity but also the value of inundation distance receives good agreements with observed values.", "the MBM result takes a longer time to reach the zero-velocity stage and the resulting inundation distance is longer than the BM result.": This is very good! However, not yet clear which

mechanism produces the real pattern. And the pattern seen in the simulation is very weak as compared with the real field event. Please discuss.

L254-255. "The sliding slope of the mudslide is developed automatically by MBM. This is big progress in studying mudslide flows.": No! This slope is also seen in BM. So, not clear what you want to say.

L314: References: Improve including suggested.

Fig. 4: Interchange panel a and b.

Fig. 6: panel d very basic, and e is in Fig. 7. So, you could remove these panels.

Fig. 7: "liquefied tailings" –> "deposited tailings". Local elevation changes in b (field photo) is very strong, however in a (simulation) it is very weak. Explain, and try to improve.

Fig. 9: Panel with time = 30 s: Not the same time, and not the geometry, this is somehow similar to the real geometric pattern in Fig. 7 b. I would expect to see similar pattern of the final geometry.

---

## Referee Comment (RC2) · Anonymous Referee #2 · 2 Jul 2020

Wu et al. present a modified rheological model for the simulation of mudflows, exploring the role of rheology in the formation of a static and a fluid region. The main highlight from this work is its three-dimensional implementation, which might come useful in a non-flat terrain and when facing obstacles. The authors focus on the role of viscosity as a key parameter for describing the kinematics of mudflows, comparing the different implementations of three rheological models (e.g., BM, CBM, MBM). Overall, the implementation is promising, but the manuscript, in its current state, does not provide a strong message for supporting the modified bi-viscosity model (MBM) as an ideal rheological representation of mudflows. My main concerns on this work are:

[Figure]

1. The validations of the numerical framework and rheological models in Sec. 3 do not include the MBM, leaving aside the comparison with the proposed ideal model for mudflows. Moreover, the numerical modifications and assumptions for adapting the model from 3D into a 2D representation are not discussed nor evident.

2. It is unclear, why the authors choose to simulate the 1966 East Texas event. If the authors interest is to highlight how the model can be used for tailing hazard assessment, then a detailed description of the event and the mobilized materials is needed. Moreover, given the frequency of tailing failures, it is tempting to see the model being validated with more cases.

3. However, if the authors motivation with the 1966 event is to prove how the MBM rheology reproduce a more accurately a mudflow, the selection of a field event of limited information makes it difficult to assess the advantages of the rheological model. Then, the selection of a benchmark case as a dam-break model seems more suitable for this purpose.

4. I got the impression that the comparisons between the three rheological models on the 1966 event are not supported by direct measurements of the material parameters of each particular model. Also, it is not clear how these parameters are obtained and calibrated. These missing information makes a critical assessment of each model difficult and leaves the reader with a qualitative similitude.

5. The manuscript goal differs slightly between line 72 and line 293. I understand that the authors explore the formation of a plug and a sheared region within the mudflow, but disagree in referring to them as solid and liquid phases, respectively.

6. It is not clear the difference between the volume fraction r and the solid concentration Cv introduced at the end of Sec. 5. A discussion on how this parameter evolves and controls the stratification process might strengthen the authors message.

7. The authors claim in line 305 that the initiation and slip surface of the mudflow is

described in their model. However, I do not find information that supports this claim, as the event simulation assumes the sudden release of the tailing material. Therefore, the conditions leading to the tailing failure are not accounted for in their model nor studied.

Given these points and the need for further simulations or a deep reevaluation of the manuscript, I recommend the rejection of this manuscript but encourage the authors to address the previous points and submit an improved version.

---

## Author Comment (AC1) · 3 Aug 2020

Response to referee #1 comments Thank you for all of your questions, suggestions, and comments. We have improved the writing of the entire article. After a long discussion with the co-authors, due to the limited information of the event, we agree that we can't reach the conclusion of stratification. Instead, we can conclude that the un-yield region of tailing material is sturdier based on the result with a larger un-yield viscosity and yield strain rate. This article has been largely modified to satisfy the standard of NHESS. We also answer your questions one by one in detail as below:

Referee #1: Title: I don't see much of the model development, rather simulations. So,

I would suggest to remove "Model development for" from the Title. If not, please justify why you want to keep it. Answer: Agree. The title is changed to "Numerical study on the failure of the gypsum tailings dam in East Texas in 1966".

Referee #1: Abstract needs to be substantially improved/re-written Answer: Agree. The abstract has been revised as below: This paper adopts Bingham and Bi-viscosity rheology models to simulate the dynamic and kinematic processes of mudslides. The rheology models are integrated into a computational fluid dynamics code, Splash3D, to solve the incompressible Navier-Stokes equations with Volume of Fluid surface tracking algorithm. The changes of the un-yield and yield phases of the mudslide material are controlled by the yield stress and yield strain rate in Bingham and Bi-viscosity models, respectively. The integrated model is carefully validated by the theoretical results and laboratory data with good agreements. This validated model is then used to study the failure of the gypsum tailings dam in East Texas in 1966. The results show that the accuracy of predicted flood distances simulated by both models is about 73% of the observation data. To improve the prediction, a fixed large viscosity is introduced to describe the un-yield behavior of tailing material. The yield strain rate is obtained by comparing the simulated flood distance to the field data. This modified Bi-viscosity model improves not only the accuracy of the flood distance to about 97% but also the accuracy of the spread width. This result suggests that the un-yield region in the modified Bi-viscosity model was sturdier than that described in the Bingham model. However, once the tailing material yields, the tailing fluid returns to the conventional Bingham liquefied material.

Referee #1: L23: Provide References, also in the following text whenever necessary Answer: Agree. The reference is added as: According to the statistics of the World Information Service on Energy (WISE), more than 130 tailings dam failures occurred from 1961 to 2019 in the world (WISE, 2016).

Referee #1: L26-28: "the information is not complete": what do you want to say, not clear. "one dimensional profiles": of what? "contradictory": with what? Please make

these aspects clear. Answer: This paragraph has been modified to be more clear as below: The rheological properties are crucial issues in a tailing flow simulation. The tailings fluid formed by mixing tailings and water is a non-Newtonian fluid in nature (Henriquez and Simms, 2009) with complex rheological properties. The travel distance and the spreading of a tailings flow are affected by the rheological equation (Yu et al., 2020). As for concerning the determination of the rheological parameters, Henriquez and Simms, (2009) determined the yield stress and viscosity of tailings flow using rheometer and slump tests. The mixture of different materials leads to a complex, yet not well understood rheological behavior (Von Boetticher et al., 2017). Field observations of mudflow behavior and rheology are challenging and still rare. Numerical modeling is chosen when an assessment of mudflow behavior is needed for planning, zoning, and hazard assessment (Scheuner et al., 2011; Christen et al., 2012; Kattel et al., 2016; Mergili et al., 2017). Most models require direct calibration to capture site-specific behavior. However, reliable calibration data are scarce, and laboratory experiments are difficult to upscale to field situations (Von Boetticher et al., 2017).

Referee #1: L32: –> (means you may consider to include, or change to): or the Coulomb viscoplastic model (Pudasaini and Mergili, 2019: https://doi.org/10.1029/2019JF005204). Answer: These sentences have been modified as below: Mudflow can be simulated by Bingham model (Schamber and MacArthur, 1985, Liu and Mei, 1989, Liu et al., 2016), Herschel–Bulkley model (Bates and Ancey, 2017; Huang and García, 1998), and the Coulomb-viscoplastic model (Pudasaini and Mergili, 2019) by incorporating them with the depth-integrated equations models.

Referee #1: L51: "with strong vertical fluid particle acceleration." –> "with strong vertical fluid particle acceleration (Domnik et al., 2013: https://doi.org/10.1016/j.jnnfm.2012.03.001, https://doi.org/10.1016/j.jnnfm.2013.07.005)." Answer: These sentences have been modified as below: The depth-integrated model is simplified from Navier-Stokes

equations by ignoring the vertical acceleration (Han et al., 2019). The depth-integrated model is suitable for predicting the flow without strong vertical acceleration or a sharp velocity shearing (Khattri and Pudasaini, 2019). The vertical acceleration and velocity shearing are important in the case of a slope with rugged topography or mudslide overtopping a structure. Inside a complex 3D flow structure, the tailings material will transfer from an un-yield/plug zone to a yield/liquefied/sheared zone if the shear stress is greater than the yield shear stress. Before the tailings material reaching this plug zone, the liquefied zone might dominate the entire flow field due to strong shear. Adopting a three-dimensional rheology model is an alternative option to fully and globally describe the strongly converging and diverging flows (Domnik et al., 2013; Yu et al., 2020). For more detailed results, the three-dimensional Navier-Stokes equations are recommended (Lee et al., 2010; Wang et al., 2016). The Navier-Stokes equations modes were utilized to study mudflow in the 1990s. Many of these studies concentrated on solving 2D problems (O'Brien et al., 1993; Assier Rzadkiewicz et al., 1997; Huang and García, 1998). With the advance of computers in the early twentieth century, the 3D Navier–Stokes equations were able to study mudflow by Smoothed Particle Hydrodynamics (SPH) method (Dai et al., 2014; Wang et al., 2016), and projection method (VonBoetticher et al., 2017; Abadie et al., 2019; Yu et al., 2020).

Referee #1: L55-60: Please improve writing. Answer: The writing of the entire article has been greatly improved.

Referee #1: and multi-phase mass flow (Pudasaini and Mergili, 2019) Answer: Thank you for your suggestion. We added it as below: For flow rheology, Bingham model (BM) has been widely used to simulate mudflows (Coussot and Proust, 1996; Liu and Mei, 1989; Mei and Yuhi, 2001), lava flows (Griffiths, 2000), landslides (McDougall and Hungr, 2004), and multi-phase mass flow (Pudasaini and Mergili, 2019).

Referee #1: L74-82: Very basic, could be removed. Answer: Agree. They are removed.

Referee #1: L82: "However, Navier-Stokes equations derived under the assumption

of Eulerian fails in describing the un-yield motion.": What does it mean? Not clear. Answer: This sentence is not clear. It is removed.

Referee #1: L86-89: Please explain how you would obtain stratification, because this is the main aspect of this MS. And, it seems to be mostly parameter fit, increase mu_A as high as you get what you want. So, what is the mechanical aspect of this paper, and how can you justify using those mu_A values. This is a critical aspect here. Please clearly justify, discuss. Answer: Thanks for the comments. After a long discussion with the co-authors, we agree that we can't reach the conclusion of stratification. Instead, we can conclude that the un-yield region of tailing material is sturdier based on the result with a larger un-yield viscosity and yield strain rate. The mechanical aspect of this paper has been clarified as below: The rheological properties of BM can be presented as (Tanner, 1982; Byron-Bird et al., 1983): $\mu(\gamma\;\dot{} )=\{âŰă(\mu\_A=\infty$ and $\gamma\;\dot{}=0$, if $\tau<\tau\_y@\mu\_B+\tau\_y/\sqrt{(1/2\;\gamma\;\dot{}\_ij{:}\gamma\;\dot{}\_ij} )$ and $\gamma\;\dot{}>0$, if $\tau\geq\tau\_y )âŤď$ (1) where $\mu\_A$ is the viscosity of the un-yield region, $\mu\_B$ is the viscosity of the yield zone, $\tau\_y$ is the yield stress, and $\gamma\;\dot{}\_y$ is the yield strain rate, $\gamma\;\dot{}\_ij=(\partial u\;\dot{}\_i)/(\partial x\_j )+(\partial u\;\dot{}\_j)/(\partial x\_i )$. The symbol $\gamma\;\dot{}$ is the second invariant of the $\gamma\;\dot{}\_ij$, which is defined as $\gamma\;\dot{}=\sqrt{(1/2\;\gamma\;\dot{}\_ij\;\gamma\;\dot{}\_ij} )$. Despite the simplicity of the BM, the fact that the stress is indeterminate in the un-yield region, which means the exact shape and location of the yield surface(s) cannot be determined (Khabazi et al., 2016). To remedy this drawback, in the present work, the conventional Bi-viscosity model (CBM) proposed by Beverly and Tanner (1992) is adopted. This idea allows a small deformation to occur in the un-yield region(s) by treating it as an extremely high viscosity fluid. In the yield region, the material is considered a Bingham fluid. This method makes it possible for the stress to be computable in the whole domain, including the un-yield region so that the location of the yield surface can be easily determined (Khabazi et al., 2016). The rheological properties of CBM can be presented as (Beverly and Tanner, 1992; Khabazi et al., 2016): $\mu(\gamma\;\dot{} )=\{Ű�(\mu\_A=\tau\_y/\gamma\;\dot{}\_y$ ,if $\gamma\;\dot{}<\gamma\;\dot{}\_y@\mu\_B+\tau\_y/\sqrt{(1/2\;\gamma\;\dot{}\_ij{:}\gamma\;\dot{}\_ij} )$ ,if $\gamma\;\dot{}\geq\gamma\;\dot{}\_y )âŤď$ (2) Mathematically speaking, when $\gamma\;\dot{}\_y$ approaches to zero, the CBM will approach to BM. By choosing $\gamma\;\dot{}\_y$ sufficiently small, we can practically replace the

un-yield region viscosity with a higher viscosity. This guarantees that a viscous solver can handle the determination of the shape and the location of the plug(s) (Khabazi et al., 2016). However, if a mud material experiences compaction or tamping processes, the material will become sturdy, and resulting in a larger $\mu\_A$. To describe the sturdy behavior in the plug zone, a larger $\mu\_A$ and a larger $\gamma$ ÌǦ_y are required. The larger $\mu\_A$, playing a role of keeping the rigid shape, and the larger $\gamma$ ÌǦ_y, indicating the material can sustain a larger deformation. This modified Bi-viscosity model (MBM) can be written as: $\mu(\gamma$ ÌǦ )={âŰă($\mu\_A>\tau\_y/\gamma$ ÌǦ_y ,if $\gamma$ ÌǦ<$\gamma$ ÌǦ_y@$\mu\_B+\tau\_y/\sqrt{}$(1/2 $\gamma$ ÌǦ_ij:$\gamma$ ÌǦ_ij ) ,if $\gamma$ ÌǦ≥$\gamma$ ÌǦ_y )âŤď (3) In this model, the yield stress $\tau\_y$ and yield viscosity $\mu\_B$ of the tailings material are exponentially dependent on material concentration (Julien, 2010). The detailed descriptions are added to Section 5.2. To present the un-yield behavior in the plug zone, $\mu\_A$ is chosen to be infinite based on the suggestions of Assier Rzadkiewicz et al., (1997); Taibi and Messelmi, (2018); Yu et al., (2020). In this paper, the infinite number of viscosity $\mu\_A$=ãĂŰ10ãĂŮˆ10 Pa s is chosen by a sensitivity analysis. The values of yield strain rate $\gamma$ ÌǦ_y are also discussed in Section 5.2. By sensitivity analysis, $\gamma$ ÌǦ_y=0.2 s-1 is adopted to illustrate the deformation in MBM.

Referee #1: L87: –> but, this could be regularized (Papanastasiou, 1987: https://doi.org/10.1122/1.549926; Domnik et al., 2013; Pudasaini and Mergili, 2019; Yu et al., 2020). Answer: Thank you for your suggestion. The context has been improved.

Referee #1: L95-101: Can you include the effect of particle concentration in the slurry viscosity as in von Boetticher et al. (2016)? Which value of mu_A would be chosen, how/why and are there any experimental/field evidence? Please discuss. Answer: The particle concentration and value of $\mu\_A$ are discussed as below: In this model, the yield stress $\tau\_y$ and yield viscosity $\mu\_B$ of the tailings material are exponentially dependent on material concentration (Julien, 2010). The detailed descriptions are added to Section 5.2. To present the un-yield behavior in the plug zone, $\mu\_A$ is chosen to be infinite based on the suggestions of Assier Rzadkiewicz et al., (1997); Taibi and

Messelmi, (2018); Yu et al., (2020). In this paper, the infinite number of viscosity $\mu\_A=ãĂŰ10ãĂŮˆ10$ Pa s is chosen by a sensitivity analysis. The values of yield strain rate $\gamma ÌĞ\_y$ are also discussed in Section 5.2. By sensitivity analysis, $\gamma ÌĞ\_y=0.2$ s-1 is adopted to illustrate the deformation in MBM.

Referee #1: Also write explicit expression for "dot(gamma):dot(gamma)", helping the reader. Answer: Thanks for your suggestion. The explicit expression for $\gamma ÌĞ$ has been added as into the manuscript. $\gamma ÌĞ\_y$ is the yield strain rate, $\gamma ÌĞ\_ij=(\partial u ÌĞ\_i)/(\partial x\_j )+(\partial u ÌĞ\_j)/(\partial x\_i )$. The symbol $\gamma ÌĞ$ is the second invariant of the $\gamma ÌĞ\_ij$, which is defined as $\gamma ÌĞ=\sqrt{(1/2 \gamma ÌĞ\_ij \gamma ÌĞ\_ij )}$.

Does (3) avoid discontinuity, or increase it? Consideration of (3) does not seem to constitute a new model as mentioned in the Title. This appears to be defined for sim-ulation purpose to try to obtain pattern seen in the field event. If not, please provide arguments. Answer: The Eq. (3) is used to introduce the discontinuity. Agree. Eq. (3) is not a new model. So, we have modified the title as: "Numerical study on the failure of the gypsum tailings dam in East Texas in 1966".

Referee #1: L108: It is a bit confusing. Sometime you talk about un-yield sometime about fluid, and now it appears that the material is dry. What type of material, and composition is considered? Further, how do you determine r? As the material density changes during motion, do you have a transport equation for r? The VOF method was also used by Yu et al. (2020) to track the interface between the tailing fluid and air. Also, improve English in the following lines. Answer: Thanks for the comments. This part has been modified as below: In the present model, the tailings fluid is treated as a single homogeneous mud material. The tailings fluid and air are assumed to be two incompressible and non-immersible fluids. The free-surface between the tailings fluid and air is tracked by the Volume of Fluid (VOF) method (Hirt and Nichols, 1981). F=1 if a numerical cell is fully occupied by the tailings fluid; 0<F<1 if the numerical cell contains both the tailings fluid and air; F=0 if the numerical cell is fully occupied by the air. The VOF equation is given by Eq. (8): $\partial F/\partial t+âĹǦ\cdot(u\_i F)=0$ (8)

Referee #1: L116-125: (i) It is not clear if you are using (6) and (7), or only (8). (ii) If (6) and (7) are used what is the advantage? (iii) Can you provide some references? Or, do you develop all these equations (3)-(11)? The readers would wonder. (iv) Can you define what is un-yield and fluid? It is completely confusing. You mention sediment and fluid, but never mention what material/composition you are considering. Are you using un-yield + fluid two-phase mixture material? Does not seem so. Answer: (i)~(iii) This part has been removed out of the manuscript because we have just published a new paper (Chu et al., 2020), which described the details of the projection method. We also present the detailed numerical algorithm here: The projection method (DeLong, 1997) was used to decouple the velocity and pressure in the Navier-Stokes equations and to solve the Poisson pressure equation (PPE). The Navier-Stokes equations were solved by the two-step projection method. The first step was the predictor step: $(\hat{\rho}^{n+1} u^* - \hat{\rho}^n u^n)/\Delta t = -\hat{L}\breve{G}\cdot(uu)^n - \hat{L}\breve{G}P^n + \hat{L}\breve{G}\cdot(\mu^{n+1}(\hat{L}\breve{G}u + \hat{L}\breve{G}^T u)^n) + \hat{\rho}^n g^n$ (1) where the superscripts n and n+1 represent the old and new time step, respectively; superscript * represents an intermediate time between time n and n+1. The density of the cell $\hat{\rho}^{n+1}$ in the cell was calculated by the volume-weight method: $\hat{\rho}^{n+1} = \hat{\rho}^{n+1}\_t$ (2) where $\_t$ is the density of the tailing fluid. Different from the conventional two-step projection method, the old-time step pressure was included in the predictor step to increase the accuracy of $u^*$. Because $u^*$ was calculated from the old-time solution explicitly, the fluid velocity $u^*$ does not satisfy the Navier-Stokes equations and needs to be corrected. The corrector step is: $(\hat{\rho}^{n+1} u^{n+1} - \hat{\rho}^{n+1} u^*)/\Delta t = -\hat{L}\breve{G}\tilde{A}\acute{U}\delta P\tilde{A}\mathring{U}^{n+1} + \hat{\rho}^{n+1} g^{n+1} - \hat{\rho}^n g^n$ (3) where the change in pressure was calculated as $\tilde{A}\acute{U}\delta P\tilde{A}\mathring{U}^{n+1} = P^{n+1} - P^n$. Taking divergence to equation (6) yields the Pressure Poisson Equation (PPE) for solving $P^{n+1}$: $\hat{L}\breve{G}\cdot(\hat{L}\breve{G}\delta P^{n+1})/\hat{\rho}^{n+1} = \hat{L}\breve{G}\cdot(u^*/\Delta t + F\_b^{n+1} - F\_b^n)$ (4) In this corrector step, the new velocity $u^{n+1}$ can be solved when the pressure $P^{n+1}$ is known. In this study, the tailing fluid is assumed to be Non-Newtonian fluids. Viscous forces are incorporated into the predictor step to estimate the cell-centered velocity field by using the * time step velocity field. This implicit approximation eliminates the limitation of the

time step for the stability criteria.

Answer: (iv) Sorry if it is confusing you. In the present model, the tailing fluid has been treated as a single homogeneous mud material. The rheology of tailing fluid is defined into two zones: un-yield/plug zone and yield/liquefied/sheared zone. The tailings material will transfer from an un-yield/plug zone to a yield/liquefied/sheared zone if the shear stress is greater than the yield shear stress, or vice versa.

Referee #1: L138: "a large viscosity parameter is imposed": again, no physical explanation. Answer: The physical explanation is added as below: The rheological properties of BM can be presented as (Tanner, 1982; Byron-Bird et al., 1983): $\mu(\gamma\,\dot{}$ )={âŰă($\mu\_A=\infty$ and $\gamma\,\dot{}$=0, if $\tau<\tau\_y@\mu\_B+\tau\_y/\sqrt{(1/2\,\gamma\,\dot{}\_ij:\gamma\,\dot{}\_ij})$ ) and $\gamma\,\dot{}$>0, if $\tau\geq\tau\_y$ )âŤď (1) where $\mu\_A$ is the viscosity of the un-yield region, $\mu\_B$ is the viscosity of the yield zone, $\tau\_y$ is the yield stress, and $\gamma\,\dot{}\_y$ is the yield strain rate, $\gamma\,\dot{}\_ij=(\partial u\,\dot{}\_i)/(\partial x\_j)+(\partial u\,\dot{}\_j)/(\partial x\_i)$. The symbol $\gamma\,\dot{}$ is the second invariant of the $\gamma\,\dot{}\_ij$, which is defined as $\gamma\,\dot{}=\sqrt{(1/2\,\gamma\,\dot{}\_ij\,\gamma\,\dot{}\_ij})$. Despite the simplicity of the BM, the fact that the stress is indeterminate in the un-yield region, which means the exact shape and location of the yield surface(s) cannot be determined (Khabazi et al., 2016). To remedy this drawback, in the present work, the conventional Bi-viscosity model (CBM) proposed by Beverly and Tanner (1992) is adopted. This idea allows a small deformation to occur in the un-yield region(s) by treating it as an extremely high viscosity fluid. In the yield region, the material is considered a Bingham fluid. This method makes it possible for the stress to be computable in the whole domain, including the un-yield region so that the location of the yield surface can be easily determined (Khabazi et al., 2016). The rheological properties of CBM can be presented as (Beverly and Tanner, 1992; Khabazi et al., 2016): $\mu(\gamma\,\dot{}$ )={âŰă($\mu\_A=\tau\_y/\gamma\,\dot{}\_y$ ,if $\gamma\,\dot{}<\gamma\,\dot{}\_y@\mu\_B+\tau\_y/\sqrt{(1/2\,\gamma\,\dot{}\_ij:\gamma\,\dot{}\_ij})$ ) ,if $\gamma\,\dot{}\geq\gamma\,\dot{}\_y$ )âŤď (2) Mathematically speaking, when $\gamma\,\dot{}\_y$ approaches to zero, the CBM will approach to BM. By choosing $\gamma\,\dot{}\_y$ sufficiently small, we can practically replace the un-yield region viscosity with a higher viscosity. This guarantees that a viscous solver can handle the determination of

the shape and the location of the plug(s) (Khabazi et al., 2016). However, if a mud material experiences compaction or tamping processes, the material will become sturdy, and resulting in a larger $\mu\_A$. To describe the sturdy behavior in the plug zone, a larger $\mu\_A$ and a larger $\gamma\,\dot{G}\_y$ are required. The larger $\mu\_A$, playing a role of keeping the rigid shape, and the larger $\gamma\,\dot{G}\_y$, indicating the material can sustain a larger deformation. This modified Bi-viscosity model (MBM) can be written as: $\mu(\gamma\,\dot{G}\,)=\{âŰă(\mu\_A>\tau\_y/\gamma\,\dot{G}\_y$ ,if $\gamma\,\dot{G}<\gamma\,\dot{G}\_y@\mu\_B+\tau\_y/\sqrt{}(1/2\,\gamma\,\dot{G}\_{ij}:\gamma\,\dot{G}\_{ij}$ ) ,if $\gamma\,\dot{G}\geq\gamma\,\dot{G}\_y$ )âŤď (3) In this model, the yield stress $\tau\_y$ and yield viscosity $\mu\_B$ of the tailings material are exponentially dependent on material concentration (Julien, 2010). The detailed descriptions are added to Section 5.2. To present the un-yield behavior in the plug zone, $\mu\_A$ is chosen to be infinite based on the suggestions of Assier Rzadkiewicz et al., (1997); Taibi and Messelmi, (2018); Yu et al., (2020). In this paper, the infinite number of viscosity $\mu\_A=ãĂŰ10ãĂŮ^10$ Pa s is chosen by a sensitivity analysis. The values of yield strain rate $\gamma\,\dot{G}\_y$ are also discussed in Section 5.2. By sensitivity analysis, $\gamma\,\dot{G}\_y=0.2$ s-1 is adopted to illustrate the deformation in MBM.

Referee #1: L158-159: Could be removed, not necessary, very basic. Table 1 is not needed, all parameters are in Fig. Answer: Thank you. Those parts have been removed.

Referee #1: L176: "Because of the yield stress, the free surface parallels to the plane bed": But not in the front. L177: "The mud front, just like a steady gravity current, eventually advances at a constant speed": Can you show it analytically, or you are talking only about numerical results? Answer: Sorry for the confusion. This part has been improved as: Because of the yield stress, the free surface needs not to be horizontal when the mud fluid is in static equilibrium, nor parallel to the plane bed when it reaches a steady-state. The mud front, like a steady gravity current, eventually advances at a constant speed with the same profile when there is a steady upstream discharge of mud (Liu and Mei, 1989). The numerical results present a similar pattern of analytical solutions to that in Liu and Mei, (1989).

Referee #1: L190: "lateral boundaries (y = 0 m and y = 400 m) are free-slip boundary": Strange! Why don't you explain why this condition is used while the base is no-slip? Answer: Thank you for the comments. This part is improved as: The downstream (x = 400 m) and lateral boundaries (y = 0 m and y = 400 m) are free-slip walls. The downstream and lateral boundaries will not affect the simulation results because the domain is set to be much larger than the predicted tailing pattern.

Referee #1: L193: The value of mu_A is huge! Can you justify it mechanically, or prove with evidence as the paper is mainly based on such values? Answer: The mechanism and the reference have been added into the manuscript as below: The rheological properties of BM can be presented as (Tanner, 1982; Byron-Bird et al., 1983): $\mu(\gamma \dot{} )=\{âŰă(\mu\_A=\infty$ and $\gamma \dot{}=0$, if $\tau< \tau\_y@\mu\_B+\tau\_y/\sqrt(1/2 \gamma \dot{}\_ij:\gamma \dot{}\_ij )$ and $\gamma \dot{}>0$, if $\tau\geq\tau\_y )âŤď$ (1) where $\mu\_A$ is the viscosity of the un-yield region, $\mu\_B$ is the viscosity of the yield zone, $\tau\_y$ is the yield stress, and $\gamma \dot{}\_y$ is the yield strain rate, $\gamma \dot{}\_ij=(\partial u \dot{}\_i)/(\partial x\_j )+(\partial u \dot{}\_j)/(\partial x\_i )$. The symbol $\gamma \dot{}$ is the second invariant of the $\gamma \dot{}\_ij$, which is defined as $\gamma \dot{}=\sqrt(1/2 \gamma \dot{}\_ij \gamma \dot{}\_ij )$. Despite the simplicity of the BM, the fact that the stress is indeterminate in the un-yield region, which means the exact shape and location of the yield surface(s) cannot be determined (Khabazi et al., 2016). To remedy this drawback, in the present work, the conventional Bi-viscosity model (CBM) proposed by Beverly and Tanner (1992) is adopted. This idea allows a small deformation to occur in the un-yield region(s) by treating it as an extremely high viscosity fluid. In the yield region, the material is considered a Bingham fluid. This method makes it possible for the stress to be computable in the whole domain, including the un-yield region so that the location of the yield surface can be easily determined (Khabazi et al., 2016). The rheological properties of CBM can be presented as (Beverly and Tanner, 1992; Khabazi et al., 2016): $\mu(\gamma \dot{} )=\{âŰă(\mu\_A=\tau\_y/\gamma \dot{}\_y$ ,if $\gamma \dot{}<\gamma \dot{}\_y@\mu\_B+\tau\_y/\sqrt(1/2 \gamma \dot{}\_ij:\gamma \dot{}\_ij )$ ,if $\gamma \dot{}\geq\gamma \dot{}\_y )âŤď$ (2) Mathematically speaking, when $\gamma \dot{}\_y$ approaches to zero, the CBM will approach to BM. By choosing $\gamma \dot{}\_y$ sufficiently small, we can practically replace the un-yield region viscosity with a higher viscosity. This guarantees that a viscous solver can handle the determination of

the shape and the location of the plug(s) (Khabazi et al., 2016). However, if a mud material experiences compaction or tamping processes, the material will become sturdy, and resulting in a larger $\mu\_A$. To describe the sturdy behavior in the plug zone, a larger $\mu\_A$ and a larger $\gamma\ \dot{G}\_y$ are required. The larger $\mu\_A$, playing a role of keeping the rigid shape, and the larger $\gamma\ \dot{G}\_y$, indicating the material can sustain a larger deformation. This modified Bi-viscosity model (MBM) can be written as: $\mu(\gamma\ \dot{G})$={âŰă($\mu\_A>\tau\_y/\gamma\ \dot{G}\_y$ ,if $\gamma\ \dot{G}<\gamma\ \dot{G}\_y$@$\mu\_B+\tau\_y/\sqrt(1/2\ \gamma\ \dot{G}\_ij:\gamma\ \dot{G}\_ij$ ) ,if $\gamma\ \dot{G}\geq\gamma\ \dot{G}\_y$ )âŤď (3) In this model, the yield stress $\tau\_y$ and yield viscosity $\mu\_B$ of the tailings material are exponentially dependent on material concentration (Julien, 2010). The detailed descriptions are added to Section 5.2. To present the un-yield behavior in the plug zone, $\mu\_A$ is chosen to be infinite based on the suggestions of Assier Rzadkiewicz et al., (1997); Taibi and Messelmi, (2018); Yu et al., (2020). In this paper, the infinite number of viscosity $\mu\_A$=ãŰ10ãŮˆ10 Pa s is chosen by a sensitivity analysis. The values of yield strain rate $\gamma\ \dot{G}\_y$ are also discussed in Section 5.2. By sensitivity analysis, $\gamma\ \dot{G}\_y$=0.2 s-1 is adopted to illustrate the deformation in MBM.

Referee #1: L196: "Splash3D": Please explain it briefly, not all the readers might know it. Answer: Thank you for your comment. The brief explanation of Splash3D is added into the manuscript as below: In this paper, the viscoplastic models, BM, CBM, and MBM are coupled with the Splash3D model. The Splash3D model is renovated from the open-source software, Truchas, which was originally developed by Los Alamos National Laboratory (Lam et al., 2007). The original program can simulate the incompressible flows with multi-fluid interfaces. The code solves three-dimensional continuity and Navier-Stokes equations by adopting the projection method (Chorin, 1968, 1997; Chu et al., 2020) and the finite volume discretization method (Eymard et al., 2000). The Splash3D model was enhanced with several hydrodynamic modules such as the Large-Eddy Simulation (LES) turbulence module (Wu, 2004; Wu and Liu, 2009), moving-solid module (Chu et al., 2020) to deal with the breaking wave, and wave-obstacle iteration problems. Readers are encouraged to read the reference Chu et al., (2020) for the detailed numerical algorithm. In this study, the Splash3D is developed with the rheological

model to solve the mudflow problems.

Referee #1: L211-215: The most important aspect of this paper is concerned with the generation of the stratification pattern observed in the field event. However, the authors could not make it quite clear what exactly is the mechanism in their model that produces these patterns. This is perhaps the major drawback of the MS. How can just gravity do that job? Please make it clear. Answer: Referred to the early reply, we can't reach the conclusion of stratification. Instead, we can conclude that the un-yield region of tailing material is sturdier based on the result with a larger un-yield viscosity and yield strain rate. Besides, several conclusions are modified in section 6. The flood distance predicted by MBM is 310 m which is closer to the filed observation with accuracy at about 97%. Not only the flood distance but also the spread width is improved in the MBM result. This result suggests that the un-yield region of tailing material in MBM results is sturdier than one in the pure Bingham material. However, once tailing material yields, the rheology returns to the Bingham properties.

Referee #1: L232-233, 250-252: "Not only freezing time, and mean velocity but also the value of inundation distance receives good agreements with observed values.", "the MBM result takes a longer time to reach the zero-velocity stage and the resulting inundation distance is longer than the BM result.": This is very good! However, not yet clear which mechanism produces the real pattern. And the pattern seen in the simulation is very weak as compared with the real field event. Please discuss. Answer: Thanks for the comments. The pattern (the white line segments) in the photo represents the horizontal displacement of the gypsum tailing. They also indicate that the velocities along the flood central streamline are faster than the other region. This part has been improved as: Fig 7 (a) shows the simulated deposited tailings from MBM. The flood distance at the freezing time t = 110 s is 310 m, which is much closer to the filed data (Jeyapalan et al., 1983). The result from MBM also shows a longer and narrower shape. However, it shall be noted that the white line segments in the aerial photo (Error! Reference source not found. (b)) are not the elevation contour lines. The white line

segments represent the horizontal displacement of the gypsum tailings. They also indicate that the velocities along the central streamline are faster than those in the other regions. An indirect validation can be seen in the free-surface velocity profile, shown in Error! Reference source not found. at t = 30 s. However, the free-surface velocity will gradually reduce to zero as the freezing time is approaching.

Referee #1: L254-255. "The sliding slope of the mudslide is developed automatically by MBM. This is big progress in studying mudslide flows.": No! This slope is also seen in BM. So, not clear what you want to say.

Answer: Sorry if it is confusing you. "The sliding slope" shall be "The slip surface". This part has been improved as: Error! Reference source not found. illustrates the strain rate profile of the initiation process of the tailing flow. The strain rate profiles in BM results show a smooth and continuous feature (Error! Reference source not found. (a)). A large amount of tailing material deforms and slides down (Error! Reference source not found. a)). On the other hands, in MBM results, the yield strain rate $\gamma$ İǦ_y=0.2 s-1 is introduced as the indicator to identify the plug and sheared zone. Because the unyield viscosity $\mu$_A=ãĂŰ10ãĂŮ^10 Pa s is much greater than $\tau$_y/$\gamma$ İǦ_y, a discontinuity pattern of the strain rate can be observed in Error! Reference source not found. (b). The yield strain rate $\gamma$ İǦ_y=0.2 s-1 keeps the plug zone rigid. The initiation process of mudslide in MBM results is different from the ones in BM results. A high strain rate appears not only near the toe of the breach but also in the gate area, which causes the sliding process and forms a slip surface. The slip surface is the interface between the un-yield and yield parts. In the bank of homogeneous mud, the slip surface of failure can be determined from the empirical method, which follows the arc of a circle that usually intersects the toe of the bank (Sun et al., 2008; Fredlund et al., 2012). However, the slip surface is developed automatically by MBM. It is worth a more profound study in the future. Error! Reference source not found. shows the strain rate profiles of BM and MBM. The slip surface (Error! Reference source not found. (b) at t = 10 s), as well as the interface between the plug/sheared zones (Fig 10 (b) and Fig 13 (b) at t = 40 s),

can be identified in the results of MBM. From the comparisons of Fig 13 (a) and (b) at t = 10 s, and also Fig (a) and (b) at t = 10 s, we can see that the slip surface is relatively sharp in the MBM results than the ones in MB.

Referee #1: L314: References: Improve including suggested. Answer: Thank you very much. The references have been included.

Referee #1: Fig. 4: Interchange panel a and b. Fig. 6: panel d very basic, and e is in Fig. 7. So, you could remove these panels. Answer: Thank you very much. Those figures have been updated as suggested.

Referee #1: Fig. 7: "liquefied tailing" –> "deposited tailing". Local elevation changes in b (field photo) is very strong, however in a (simulation) it is very weak. Explain, and try to improve. Answer: Thanks for the comments. The word "liquefied tailings" was adopted directly based on the reference Jeyapalan et al., (1983). The aerial photo of the FGT66 has been updated to a clear one. In the photo, the pattern of white lines indicates the moving speed of the moving tailings before the freezing time. However, the pattern presented in the numerical simulation is the free-surface elevation. The result from MBM matches the observation well in terms of the flood distance, flood speed, freezing time, and the boundary of the flooded area.

Referee #1: Fig. 9: Panel with time = 30 s: Not the same time, and not the geometry, this is somehow similar to the real geometric pattern in Fig. 7 b. I would expect to see similar pattern of the final geometry. Answer: The white line segments shown in the aerial photo indicates the deformation of the tailing. It also recorded the velocity during the flooding process. Fig 9 at t = 30 s also shows the velocity profile of the free-surface from MBM. This result indirectly matches the field observation in terms of the fast-moving speed at the centerline area. However, the free-surface velocity will gradually reduce to zero as the freezing time is approaching. The white line segments in the aerial photo are not the elevation contours. It is difficult to extract the elevation profile directly from the aerial photo. In this paper, the flood distance and the flood area

are chosen for the numerical comparison. They are relatively easier to be observed.

References Abadie, S., Paris, A., Ata, R., LeRoy, S., Arnaud, G., Poupardin, A., Clous, L., Heinrich, P., Harris, J., Pedreros, R. and Krien, Y.: La Palma landslide tsunami: computation of the tsunami source with a calibrated multi-fluid Navier-Stokes model and wave impact assessment with propagation models of different types, Natural Hazards and Earth System Sciences Discussions, doi:10.5194/nhess-2019-225, 2019. Assier Rzadkiewicz, S., Mariotti, C. and Heinrich, P.: Numerical simulation of submarine landslides and their hydraulic effects, Journal of Waterway, Port, Coastal and Ocean Engineering, doi:10.1061/(asce)0733-950x(1997)123:4(149), 1997. Bates, B. M. and Ancey, C.: The dam-break problem for eroding viscoplastic fluids, Journal of Non-Newtonian Fluid Mechanics, doi:10.1016/j.jnnfm.2017.01.009, 2017. Beverly, C. R. and Tanner, R. I.: Numerical analysis of three-dimensional Bingham plastic flow, Journal of Non-Newtonian Fluid Mechanics, 42(1–2), 85–115, doi:10.1016/0377-0257(92)80006-J, 1992. VonBoetticher, A., Turowski, J. M., McArdell, B. W., Rickenmann, D., Hürlimann, M., Scheidl, C. and Kirchner, J. W.: DebrisInterMixing-2.3: A finite volume solver for three-dimensional debris-flow simulations with two calibration parameters - Part 2: Model validation with experiments, Geoscientific Model Development, 10(11), 3963–3978, doi:10.5194/gmd-10-3963-2017, 2017. Byron-Bird, R., Dai, G. C. and Yarusso, B. J.: Rheology and flow of viscoplastic materials, Reviews in Chemical Engineering, doi:10.1515/revce-1983-0102, 1983. Chorin, A. J.: Numerical Solution of the Navier-Stokes Equations, Mathematics of Computation, doi:10.2307/2004575, 1968. Chorin, A. J.: A numerical method for solving incompressible viscous flow problems, Journal of Computational Physics, doi:10.1006/jcph.1997.5716, 1997. Christen, M., Bühler, Y., Bartelt, P., Leine, R., Glover, J., Schweizer, A., Graf, C., Mcardell, B. W., Gerber, W., Deubelbeiss, Y. and Feistl, T.: Integral hazard management using a unified software environment. Numerical simulation tool "RAMMS" for gravitational natural hazards, Proceedings Vol 1 12th Congress INTERPRAEVENT 2012 Grenoble France 23th 26th April 2012, doi:ISBN 978-3-901164-19-4, 2012. Chu, C. R., Wu, T. R., Tu, Y. F., Hu, S. K. and Chiu, C. L.: Interaction of two free-falling

spheres in water, Physics of Fluids, 32(3), doi:10.1063/1.5130467, 2020. Coussot, P. and Proust, S.: Slow, unconfined spreading of a mudflow, Journal of Geophysical Research: Solid Earth, doi:10.1029/96jb02486, 1996. Dai, Z., Huang, Y., Cheng, H. and Xu, Q.: 3D numerical modeling using smoothed particle hydrodynamics of flow-like landslide propagation triggered by the 2008 Wenchuan earthquake, Engineering Geology, doi:10.1016/j.enggeo.2014.03.018, 2014. Domnik, B., Pudasaini, S. P., Katzenbach, R. and Miller, S. A.: Coupling of full two-dimensional and depth-averaged models for granular flows, Journal of Non-Newtonian Fluid Mechanics, 201, 56–68, doi:10.1016/j.jnnfm.2013.07.005, 2013. Eymard, R., Gallouët, T. and Herbin, R.: Finite volume methods, Handbook of Numerical Analysis, doi:10.1016/S1570-8659(00)07005-8, 2000. Griffiths, R. W.: The Dynamics of Lava Flows, Annual Review of Fluid Mechanics, doi:10.1146/annurev.fluid.32.1.477, 2000. Han, Z., Su, B., Li, Y., Wang, W., Wang, W., Huang, J. and Chen, G.: Numerical simulation of debris-flow behavior based on the SPH method incorporating the Herschel-Bulkley-Papanastasiou rheology model, Engineering Geology, doi:10.1016/j.enggeo.2019.04.013, 2019. Henriquez, J. and Simms, P.: Dynamic imaging and modelling of multilayer deposition of gold paste tailings, Minerals Engineering, doi:10.1016/j.mineng.2008.05.010, 2009. Hirt, C. W. and Nichols, B. D.: Volume of fluid (VOF) method for the dynamics of free boundaries, Journal of Computational Physics, doi:10.1016/0021-9991(81)90145-5, 1981. Huang, X. and García, M. H.: A Herschel-Bulkley model for mud flow down a slope, Journal of Fluid Mechanics, doi:10.1017/S0022112098002845, 1998. Jeyapalan, J. K., Duncan, J. M. and Seed, H. B.: Investigation of flow failures of tailings dams, Journal of Geotechnical Engineering, doi:10.1061/(ASCE)0733-9410(1983)109:2(172), 1983. Julien, P. Y.: Erosion and sedimentation, Second edition., 2010. Kattel, P., Khattri, K. B., Pokhrel, P. R., Kafle, J., Tuladhar, B. M. and Pudasaini, S. P.: Simulating glacial lake outburst floods with a two-phase mass flow model, Annals of Glaciology, doi:10.3189/2016AoG71A039, 2016. Khabazi, N. P., Taghavi, S. M. and Sadeghy, K.: Peristaltic flow of Bingham fluids at large Reynolds numbers: A numerical study, Journal of Non-Newtonian Fluid Mechanics, 227, 30–44, doi:10.1016/j.jnnfm.2015.11.004, 2016. Khattri, K. B. and Pudasaini, S. P.: Channel flow simulation of a mixture with a full-dimensional generalized quasi two-phase model, Mathematics and Computers in Simulation, 165, 280–305, doi:10.1016/j.matcom.2019.03.014, 2019. Lee, E. S., Violeau, D., Issa, R. and Ploix, S.: Application of weakly compressible and truly incompressible SPH to 3-D water collapse in waterworks, Journal of Hydraulic Research, doi:10.1080/00221686.2010.9641245, 2010. Liu, K. F. and Mei, C. C.: Slow spreading of a sheet of Bingham fluid on an inclined plane, Journal of Fluid Mechanics, doi:10.1017/S0022112089002685, 1989. Liu, Y., Balmforth, N. J., Hormozi, S. and Hewitt, D. R.: Two–dimensional viscoplastic dambreaks, Journal of Non-Newtonian Fluid Mechanics, doi:10.1016/j.jnnfm.2016.05.008, 2016. McDougall, S. and Hungr, O.: A model for the analysis of rapid landslide motion across three-dimensional terrain, Canadian Geotechnical Journal, doi:10.1139/T04-052, 2004. Mei, C. C. and Yuhi, M.: Slow flow of a Bingham fluid in a shallow channel of finite width, Journal of Fluid Mechanics, doi:10.1017/S0022112000003013, 2001. Mergili, M., Fischer, J. T., Krenn, J. and Pudasaini, S. P.: R.avaflow v1, an advanced open-source computational framework for the propagation and interaction of two-phase mass flows, Geoscientific Model Development, doi:10.5194/gmd-10-553-2017, 2017. O'Brien, J. S., Julien, P. Y. and Fullerton, W. T.: Two-dimensional water flood and mudflow simulation, Journal of Hydraulic Engineering, doi:10.1061/(ASCE)0733-9429(1993)119:2(244), 1993. Pudasaini, S. P. and Mergili, M.: A Multi-Phase Mass Flow Model, Journal of Geophysical Research: Earth Surface, 124(12), 2920–2942, doi:10.1029/2019JF005204, 2019. Schamber, D. R. and MacArthur, R. C.: One-dimensional model for mud flows., 1985. Scheuner, T., Schwab, S. and McArdell, B. W.: Application of a two-dimensional numerical model in risk and hazard assessment in Switzerland, in International Conference on Debris-Flow Hazards Mitigation: Mechanics, Prediction, and Assessment, Proceedings., 2011. Taibi, H. and Messelmi, F.: Effect of yield stress on the behavior of rigid zones during the laminar flow of Herschel-Bulkley fluid, Alexandria Engineering Journal, doi:10.1016/j.aej.2017.01.001, 2018. Tanner,

[Figure]

R. I.: Engineering rheology, CEA, Chemical Engineering in Australia, 1982. Wang, W., Chen, G., Han, Z., Zhou, S., Zhang, H. and Jing, P.: 3D numerical simulation of debris-flow motion using SPH method incorporating non-Newtonian fluid behavior, Natural Hazards, doi:10.1007/s11069-016-2171-x, 2016. WISE: Chronology of major tailings dam failures, WISE Uranium Project-Tailings Dam Safety, 17 [online] Available from: http://www.wise-uranium.org/mdaf.html, 2016. Wu, T.-R.: A Numerical Study of Three-Dimensional Breaking Waves and Turbulence Effects All Rights Reserved., 2004. Wu, T. R. and Liu, P. L. F.: Numerical study on the three-dimensional dambreak bore interacting with a square cylinder, in Selected Papers of the Symposium Held in Honor of Philip L-F Liu's 60th Birthday - Nonlinear Wave Dynamics., 2009. Yu, D., Tang, L. and Chen, C.: Three-dimensional numerical simulation of mud flow from a tailing dam failure across complex terrain, Natural Hazards and Earth System Sciences, 20(3), 727–741, doi:10.5194/nhess-20-727-2020, 2020.

Please also note the supplement to this comment:
https://nhess.copernicus.org/preprints/nhess-2020-126/nhess-2020-126-AC1-supplement.pdf

---

## Author Comment (AC2) · 3 Aug 2020

Response to referee #2 comments We sincerely thank you for all of your questions, suggestions, and comments. They are really useful for us to improve our manuscript. According to your comments and a long discussion with co-authors, we decide to change the title, the abstract, as well as the main conclusions. The writing of the entire article has been largely improved to satisfy the standard of NHESS.

The validations of the numerical framework and rheological models in Sec. 3 do not include the MBM, leaving aside the comparison with the proposed ideal model for mud-flows. Moreover, the numerical modifications and assumptions for adapting the model

from 3D into a 2D representation are not discussed nor evident. Answer: Thanks for the comments. The validation framework is: We validate the accuracy of the Bingham model (BM) by two cases. After the validations, the BM and the conventional Bi-viscous model (CBM) are used to simulated the event of FGT66. The sensitive analysis shows that the results of BM and CBM are nearly identical when the yield strain rate is small. In the end, a large viscosity of plug zone is proposed in the modified Bi-viscosity model (MBM) as the suggestion of Assier Rzadkiewicz et al., 1997; Taibi and Messelmi, 2018; Yu et al., 2020 to describe the sturdy behavior in the un-yield region. The only difference between CBM and MBM is the material parameters. The numerical code is kept the same. Therefore, the code validation for MBM could be referred to as the CBM. The numerical code remains the same for a 2D and 3D problem. 2D problem is one special case of the 3D problem as the free-slip boundary conditions are applied to the lateral boundaries.

It is unclear, why the authors choose to simulate the 1966 East Texas event. If the authors interest is to highlight how the model can be used for tailing hazard assessment, then a detailed description of the event and the mobilized materials is needed. Moreover, given the frequency of tailing failures, it is tempting to see the model being validated with more cases. Answer: The purpose of this study is to give a flexibility for illustrating the sturdy un-yield behavior numerically in the mudflow by migrating BM to CBM, and from CBM to MBM. Because of the clear setup and simplicity in geometry and topography, the event of FGT66 is chosen and discussed.

However, if the authors motivation with the 1966 event is to prove how the MBM rheology reproduce a more accurately a mudflow, the selection of a field event of limited information makes it difficult to assess the advantages of the rheological model. Then, the selection of a benchmark case as a dam-break model seems more suitable for this purpose. Answer: Thanks for your comments. We agree that choosing a benchmark case as a dam-break model will be a better choice. However, in our limited knowledge, no experiments have been done for MBM. For the FGT66, the geometry, and fundamental material parameters were reported in Jeyapalan et al., (1983); Pastor et al., (2002); Chen and Peng, (2006).

I got the impression that the comparisons between the three rheological models on the 1966 event are not supported by direct measurements of the material parameters of each particular model. Also, it is not clear how these parameters are obtained and calibrated. These missing information makes a critical assessment of each model difficult and leaves the reader with a qualitative similitude. Answer: Thanks for the comments. The material parameters are obtained from the publications such as Jeyapalan et al., (1983); Pastor et al., (2002); Chen and Peng, (2006). More information is added to the manuscript: Based on the parameters reported by Jeyapalan et al., 1983, Pastor et al., (2002), and Chen and Peng, (2006), the yield stress of the tailings is $\tau$_y=ãĂŰ10ãĂŮˆ3 Pa, the viscosity of the liquefied zone is $\mu$_B=50 Pa s, and the density is =1400 kg m-3. The viscosity of the plug zone is suggested to be infinite (e.g. $\mu$_A=ãĂŰ10ãĂŮˆ10 Pa s) by Assier Rzadkiewicz et al., (1997); Taibi and Messelmi, (2018); Yu et al., (2020). In this model, the yield stress $\tau$_y and yield viscosity $\mu$_B of the tailings material are exponentially dependent on material concentration (Julien, 2010). The detailed descriptions are added to Section 5.2. To present the un-yield behavior in the plug zone, $\mu$_A is chosen to be infinite based on the suggestions of Assier Rzadkiewicz et al., (1997); Taibi and Messelmi, (2018); Yu et al., (2020). In this paper, the infinite number of viscosity $\mu$_A=ãĂŰ10ãĂŮˆ10 Pa s is chosen by a sensitivity analysis. The values of yield strain rate $\gamma$ ÌĞ_y are also discussed in Section 5.2. By sensitivity analysis, $\gamma$ ÌĞ_y=0.2 s-1 is adopted to illustrate the deformation in MBM.

The manuscript goal differs slightly between line 72 and line 293. I understand that the authors explore the formation of a plug and a sheared region within the mudflow, but disagree in referring to them as solid and liquid phases, respectively. Answer: Thanks for the comments. The solid phase has been changed to the un-yield phase, and the liquefied phase has been changed to the yield phase.

It is not clear the difference between the volume fraction r and the solid concentration

Cv introduced at the end of Sec. 5. A discussion on how this parameter evolves and controls the stratification process might strengthen the authors message. Answer: The concentration Cv is used to determine the yield stress $\tau\_0$ and the yield viscosity $\mu\_B$ of the mud material (Julien, 2010). The volume fraction of mud, F (has been changed from r), in the VOF equation is used to track the mud free-surface. The detailed algorithm of the VOF method can be found in the paper, we recently published (Chu et al., 2020).

The authors claim in line 305 that the initiation and slip surface of the mudflow is described in their model. However, I do not find information that supports this claim, as the event simulation assumes the sudden release of the tailing material. Therefore, the conditions leading to the tailing failure are not accounted for in their model nor studied. Answer: Thanks for the comments. This part has been improved as: Error! Reference source not found. illustrates the strain rate profile of the initiation process of the tailing flow. The strain rate profiles in BM results show a smooth and continuous feature (Error! Reference source not found. (a)). A large amount of tailing material deforms and slides down (Error! Reference source not found. a)). On the other hands, in MBM results, the yield strain rate $\gamma$ ÌĞ_y=0.2 s-1 is introduced as the indicator to identify the plug and sheared zone. Because the un-yield viscosity $\mu\_A$=ãĂŰ10ãĂŮˆ10 Pa s is much greater than $\tau\_y/\gamma$ ÌĞ_y, a discontinuity pattern of the strain rate can be observed in Error! Reference source not found. (b). The yield strain rate $\gamma$ ÌĞ_y=0.2 s-1 keeps the plug zone rigid. The initiation process of mudslide in MBM results is different from the ones in BM results. A high strain rate appears not only near the toe of the breach but also in the gate area, which causes the sliding process and forms a slip surface. The slip surface is the interface between the un-yield and yield parts. In the bank of homogeneous mud, the slip surface of failure can be determined from the empirical method, which follows the arc of a circle that usually intersects the toe of the bank (Sun et al., 2008; Fredlund et al., 2012). However, the slip surface is developed automatically by MBM. It is worth a more profound study in the future. Error! Reference source not found. shows the strain rate profiles of BM and MBM. The slip surface (Error! Reference source not found. (b) at t = 10 s), as well as the interface between

the plug/sheared zones (Fig 10 (b) and Fig 13 (b) at t = 40 s), can be identified in the results of MBM. From the comparisons of Fig 13 (a) and (b) at t = 10 s, and also Fig (a) and (b) at t = 10 s, we can see that the slip surface is relatively sharp in the MBM results than the ones in MB.

References Assier Rzadkiewicz, S., Mariotti, C. and Heinrich, P.: Numerical simulation of submarine landslides and their hydraulic effects, Journal of Waterway, Port, Coastal and Ocean Engineering, 123(4), 149–157, doi:10.1061/(asce)0733-950x(1997)123:4(149), 1997. Chen, S. C. and Peng, S. H.: Two-dimensional numerical model of two-layer shallow water equations for confluence simulation, Advances in Water Resources, doi:10.1016/j.advwatres.2005.12.001, 2006. Jeyapalan, J. K., Duncan, J. M. and Seed, H. B.: Analyses of flow failures of mine tailings dams, Journal of Geotechnical Engineering, doi:10.1061/(ASCE)0733-9410(1983)109:2(150), 1983a. Jeyapalan, J. K., Duncan, J. M. and Seed, H. B.: Investigation of flow failures of tailings dams, Journal of Geotechnical Engineering, doi:10.1061/(ASCE)0733-9410(1983)109:2(172), 1983b. Julien, P. Y.: Erosion and sedimentation, Second edition., 2010. Pastor, M., Quecedo, M., Fernádez Merodo, J. A., Herrores, M. I., González, E. and Mira, P.: Modelling tailings dams and mine waste dumps failures, Geotechnique, 52(8), 579–591, doi:10.1680/geot.2002.52.8.579, 2002. Taibi, H. and Messelmi, F.: Effect of yield stress on the behavior of rigid zones during the laminar flow of Herschel-Bulkley fluid, Alexandria Engineering Journal, doi:10.1016/j.aej.2017.01.001, 2018. Yu, D., Tang, L. and Chen, C.: Three-dimensional numerical simulation of mud flow from a tailing dam failure across complex terrain, Natural Hazards and Earth System Sciences, 20(3), 727–741, doi:10.5194/nhess-20-727-2020, 2020.

Please also note the supplement to this comment:
https://nhess.copernicus.org/preprints/nhess-2020-126/nhess-2020-126-AC2-supplement.pdf